# Different theta frameworks coexist in the rat hippocampus and are coordinated during memory-guided and novelty tasks

**Víctor J López-Madrona[1], Elena Pérez-Montoyo[1], Efrén Álvarez-Salvado[1], David Moratal[2], Oscar Herreras[3], Ernesto Pereda[4,5], Claudio R Mirasso[6], Santiago Canals[1]***

[1]Instituto de Neurociencias, Consejo Superior de Investigaciones Científicas, Universidad Miguel Hernández, San Juan de Alicante, Spain; [2]Centro de Biomateriales e Ingeniería Tisular, Universitat Politècnica de València, Valencia, Spain; [3]Instituto Cajal, Consejo Superior de Investigaciones Científicas, Madrid, Spain; [4]Departamento de Ingeniería Industrial & IUNE, Escuela Superior de Ingeniería y Tecnología, Universidad de La Laguna, La Laguna, Spain; [5]Laboratory of Cognitive and Computational Neuroscience, Center for Biomedical Technology, Madrid, Spain; [6]Instituto de Física Interdisciplinar y Sistemas Complejos, IFISC (CSIC-UIB), Campus Universitat de les Illes Balears, Palma de Mallorca, Spain

**Abstract** Hippocampal firing is organized in theta sequences controlled by internal memory processes and by external sensory cues, but how these computations are coordinated is not fully understood. Although theta activity is commonly studied as a unique coherent oscillation, it is the result of complex interactions between different rhythm generators. Here, by separating hippocampal theta activity in three different current generators, we found epochs with variable theta frequency and phase coupling, suggesting flexible interactions between theta generators. We found that epochs of highly synchronized theta rhythmicity preferentially occurred during behavioral tasks requiring coordination between internal memory representations and incoming sensory information. In addition, we found that gamma oscillations were associated with specific theta generators and the strength of theta-gamma coupling predicted the synchronization between theta generators. We propose a mechanism for segregating or integrating hippocampal computations based on the flexible coordination of different theta frameworks to accommodate the cognitive needs.

**\*For correspondence:**
scanals@umh.es

**Competing interests:** The authors declare that no competing interests exist.

## Introduction

The hippocampal formation flexibly combines computations subserving spatial navigation, driven by external environmental cue stimuli (*McNaughton et al., 1983*; *O'Keefe and Nadel, 1978*), but also memory processing, dependent on internally generated firing sequences (*Pastalkova et al., 2008*; *Wang et al., 2015*). The characteristic oscillatory activity patterns in brain networks have been proposed as a mechanism to organize different computations and, depending on the cognitive needs, integrate or segregated them in oscillatory cycles (*Buzsáki and Draguhn, 2004*; *Engel et al., 2001*; *Lisman and Jensen, 2013*). In the hippocampus, theta and gamma oscillations are the most prominent rhythms recorded in freely moving animals (*Buzsáki, 2002*; *Colgin, 2016*; *Colgin, 2013*; *Vanderwolf, 1969*). Theoretical and experimental work in the hippocampus have associated the processing of environmental cues and the encoding of memories with the input from the entorhinal cortex (EC) arriving at CA1 at a particular phase of the theta cycle, and the retrieval of memories with the CA3 output to CA1 in a different phase of the cycle (*Douchamps et al., 2013*;

**eLife digest** In the brain, a vast number of neurons coordinate their activity to support complex cognitive processes. One of the best places to see this in action is the hippocampus, a brain structure with a key role in memory and navigation.

The hippocampus shows waves of electrical activity, which represent the synchronized firing of large numbers of neurons. The hippocampus can generate multiple rhythms at once. The two main rhythms are theta and gamma. Theta waves are slow, with a frequency of about 8 Hertz. Gamma waves are faster with a frequency of up to 120 Hertz or even more.

Theta waves are always present in the brains of freely moving animals, whereas gamma waves occur in brief bursts. These bursts usually correspond to a particular point on the theta wave. One burst may occur just before each peak of the theta wave, for example, whereas another burst may occur just after the peak. This separation enables individual bursts of gamma to carry different messages without them becoming mixed up. This is similar to how radio stations broadcast their signals at different carrier frequencies to avoid interference.

By recording hippocampal activity in rats exploring a maze, Lopez-Madrona et al. now show that the hippocampus has not one, but three generators of theta waves. Having three sources of theta, each of which can be synchronized with gamma, provides a more versatile system for encoding and sending information. It also means that the three theta generators can vary the degree to which they coordinate their firing. This helps the brain combine or separate streams of information as required. By working together to create a single theta rhythm, for example, the three theta generators can help animals combine information stored in memory with incoming sensory input.

How the coordination of theta rhythms in the hippocampus influences the activity of other brain regions involved in learning and memory remains unclear. However, uncoupling of theta and gamma waves seems to be an early sign of Alzheimer's disease and can also be seen in the brains of people with schizophrenia and other psychiatric disorders. Understanding how this process occurs could provide clues to the origin of these disorders.

*Hasselmo et al., 2002*; *Siegle and Wilson, 2014*). Information transmission between these regions is proposed to occur in gamma oscillations of different frequencies organized in the phase of the slower CA1 theta rhythm (*Colgin et al., 2009*; *Lisman and Idiart, 1995*; *Lisman and Jensen, 2013*). These theta-gamma associations, known as cross-frequency coupling (CFC), are modulated during exploration and memory-guided behaviors (*Cabral et al., 2014*; *Canolty et al., 2006*; *Colgin, 2015*; *Fernández-Ruiz et al., 2017*; *Lasztóczi and Klausberger, 2016*; *Lasztóczi and Klausberger, 2014*; *Schomburg et al., 2014*; *Tort et al., 2008*). Recent studies have further shown that theta-gamma interaction may vary in a cycle-by-cycle manner within a global hippocampal theta rhythmicity (*Dvorak et al., 2018*; *Lopes-Dos-Santos et al., 2018*; *Zhang et al., 2019*). However, theta oscillations originating in different anatomical layers of the hippocampus are known to coexist (*Alonso and García-Austt, 1987*; *Bland and Whishaw, 1976*; *Buzsáki, 2002*; *Charpak et al., 1995*; *Green and Rawlins, 1979*; *Vanderwolf C et al., 1973*; *Kramis et al., 1975*; *Montgomery et al., 2009*; *Vanderwolf, 1969*; *Winson, 1974*) and, therefore, theta-gamma interactions need to be interpreted in the context of multiple rhythm generators.

In addition to the classical medial septum/diagonal band of Broca input imposing a global rhythmicity to the hippocampus and EC, important rhythm generators are located in EC layers II (EC2) and III (EC3), whose activity reach the dentate gyrus (DG) and hippocampus proper through the perforant and temporoammonic pathways, respectively, and from CA3 activity reaching CA1 *stratum radiatum* through the Schafer collaterals (*Buzsáki, 2002*). Importantly, although theta oscillations in the hippocampus are most commonly studied as a unique coherent oscillation across hippocampal layers, exhibiting a characteristic amplitude/phase vs. depth variation (*Buzsáki, 2002*), the frequency and phase of the CA3 theta rhythm generator was shown to change relatively independently from the EC theta inputs (*Kocsis et al., 1999*; *Montgomery et al., 2009*). How these multiple theta rhythm generators and pathway-specific gamma oscillations interact in the hippocampus is not well understood. One appealing possibility is that different theta oscillations may represent different

theta-gamma coding frameworks, providing the substrate to segregate, but also to integrate computations.

Here we investigated the function of pathway-specific synchronization of oscillatory activity in the hippocampus of rats freely exploring known and novel environments and resolving a T-maze. Using high density electrophysiological recordings aided by source separation techniques we characterized the dynamic properties of three different theta and three different gamma dipoles in the hippocampus with origins in the CA3 Schaffer collateral layer, the EC3 projection to the *stratum lacunosum-moleculare* and the EC2 projection to the mid-molecular layer of the DG, respectively, and found strong support for the existence of different theta-gamma frameworks. Using optogenetic tools targeted to CA3 parvalbumin interneurons, we show the specific modulation of the CA3-associated *vs.* the EC-associated theta generators, demonstrating independent theta oscillations in the hippocampus. Nevertheless, phase shifts between the identified theta frameworks served to coordinate them in pairs or triads, in a sub-second timescale. We then characterized theta-gamma interactions between the different pathways and established an association with the synchronization state in the hippocampal network. Theta-gamma CFC was stronger during higher theta synchronization and we found that pathway-specific gamma oscillations consistently precede theta phase shifts. Finally, we investigated the functional role of these theta-gamma frameworks for contextual learning requiring the update of an existent memory with the changes found in the environment. We found that both, theta-gamma CFC and the coordination between theta oscillations, were consistently higher during mismatch novelty and memory guided decisions, in situations in which the representation of the context from memory is compared against the incoming sensory information.

## Results

### Pathway-specific theta and gamma oscillations

We performed electrophysiological recordings using linear array electrodes across the dorsal hippocampus in five rats (see Materials and methods, *Figure 1*). Recordings were carried out while the animal freely explored a familiar open field (Figures 1, 2, 3, 4, 5, 6), a novel open field (Figure 6) or a T-maze (Figure 7). Using spatial discrimination techniques to separate LFP sources contributed by different synaptic pathways, based on independent component analysis (ICA, *Figure 1—figure supplement 1*; *Benito et al., 2014*; *Fernández-Ruiz and Herreras, 2013*; *Herreras, 2016*; *Herreras et al., 2015*; *Lęski et al., 2010*; *Makarov et al., 2010*; *Makarova et al., 2011*; *Ortuño et al., 2019*; *Schomburg et al., 2014*), we dissected three robust components in all subjects (*Figure 1*; Materials and methods). The maximum voltages (*Figure 1b*) and dipoles in the current source density (CSD) depth profiles (*Figure 1c*) of the three components matched the stratified distribution of known terminal fields in the hippocampus and the currents resulting from stimulation of the corresponding pathways, as previously shown (*Benito et al., 2014*; *Fernández-Ruiz and Herreras, 2013*; *Lasztóczi and Klausberger, 2014*; *Schomburg et al., 2014*). The first component was located in the *stratum radiatum*, where the CA3 Schaffer collateral/commissural pathway targets the CA1 region (labelled as Schaffer component or Sch-IC). The second matched the EC3 projection in the *stratum lacunosum-moleculare* (lm-IC), and the third one the perforant pathway from EC2 to the mid-molecular layer of the DG (PP-IC). These three components, referred to as pathway-specific LFPs or IC-LFPs, represent the synaptic contributions with distinct anatomical origins recorded in the LFP (*Herreras, 2016*).

The power spectra of these signals exhibited a clear peak at theta frequency (6–10 Hz) and broadband gamma activity (*Figure 1e*). CA3 and EC3 neurons have been shown to fire phase locked to discrete gamma band oscillations in the downstream Sch-IC and lm-IC, respectively (*Fernández-Ruiz et al., 2012a*; *Schomburg et al., 2014*), with gamma oscillations segregated in the phases of the theta wave recorded in CA1 (*Colgin et al., 2009*; *Lasztóczi and Klausberger, 2014*; *Schomburg et al., 2014*). In good agreement, pathway-specific gamma activities were distributed in the theta cycle, with lm-IC close to the theta peak (π radians) and followed by Sch-IC (*Figure 1f*), showing consistency with the firing properties of principal neurons in their respective upstream afferent layers. Similarly, entorhinal principal cells in EC2 and EC3 were shown to fire in anti-phase, relative to the theta oscillation (*Mizuseki et al., 2009*) and, accordingly, large amplitude gamma oscillations in PP-IC and lm-IC in our recordings were found shifted 180° (*Figure 1f*). These results

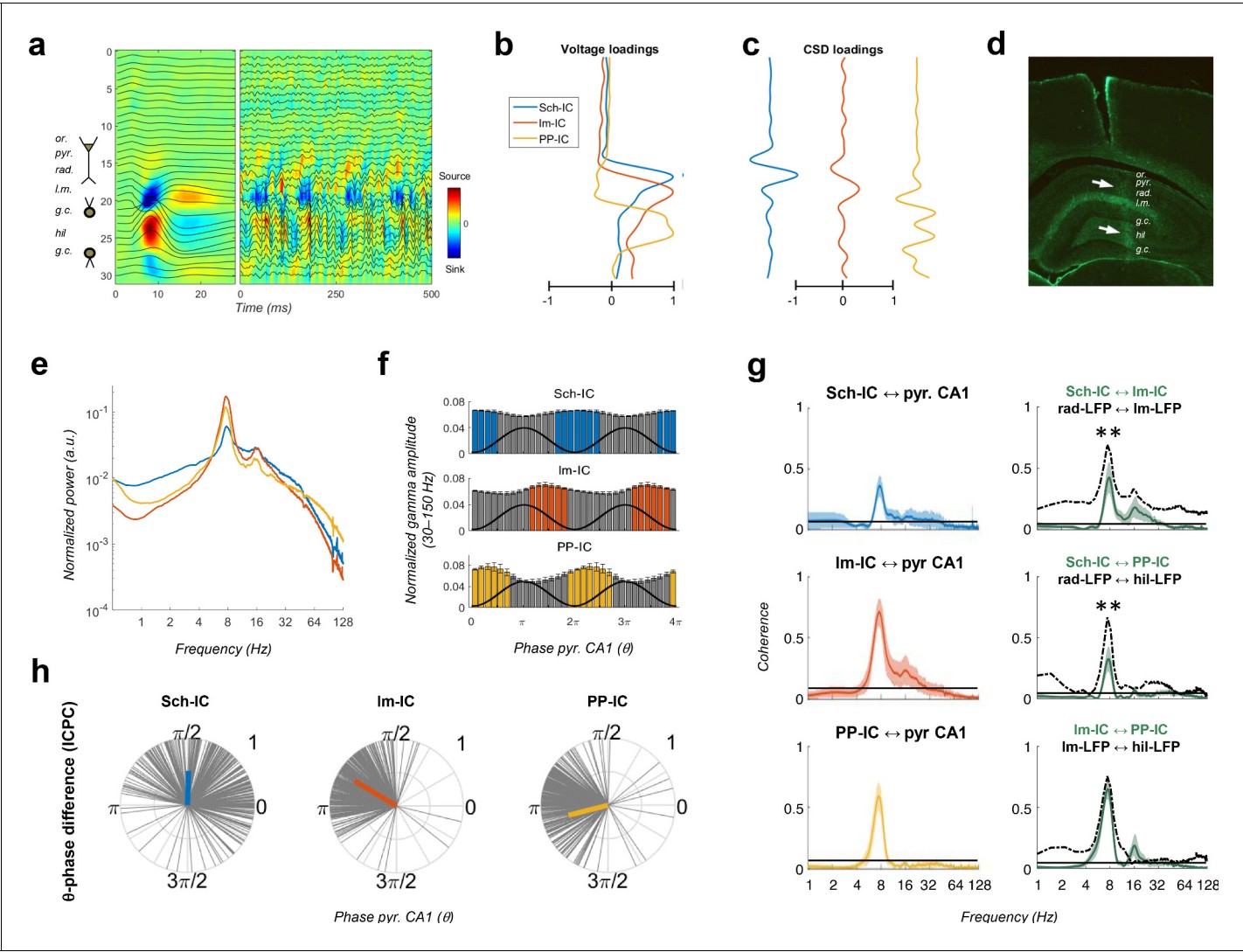

**Figure 1.** Pathway-specific local field potentials (LFPs). (a) Depth profiles of the electrophysiological signals recorded in the dorsal hippocampus (32 recordings, sites spaced every 100 µm) evoked by an electric pulse stimulating the perforant pathway (left panel) or during resting activity (right panel). Black traces represent the LFPs and color maps the corresponding CSD. Evoked activity was used to consistently localize the electrodes during implantation. Or, *stratum oriens*; pyr, pyramidal layer; rad, *stratum radiatum*; lm, *stratum lacunosum-moleculare*; gc, granule cell layer; hil, hilus. (b and c) Examples of voltage- and CSD-loadings of the three pathway-specific LFPs extracted with the ICA, with maximum loadings overlapping the corresponding afferent layers in the *str. radiatum* (Sch-IC), *lacunosum-moleculare* (lm-IC) and the molecular layer of the dentate gyrus (PP-IC). (d) Position of the recording electrode (arrows) in one representative animal. The histological section is immunostained with GFAP antibodies. (e) The power spectra of the three IC-LFPs averaged across subjects show a clear peak at the theta frequency and a broadband gamma activity. (f) Distribution of gamma amplitude (mean ± s.e.m.) in IC-LFPs along the theta cycles of the LFP recorded in pyr CA1, where its trough and peak coincide with 0 and π radians, respectively. Black waves are an example of the theta oscillation. Color coded bars represent statistically significant values relative to the surrogate distribution in all animals (Materials and methods). (g) On the left panel, we plot the results of the coherence analysis comparing the IC-LFPs against the LFP recorded in pyr CA1. Black lines represent the statistical threshold. On the right panel, the coherence analysis between IC-LFPs (green lines) is compared to the coherence between raw LFP channels recorded at the layers with maximum contribution to each IC-LFP (dashed lines, **p<0.01, we used a two-way ANOVA to compare the theta coherence between pairs of IC-LFPs versus theta coherence between raw LFPs, followed by Bonferroni correction, F(1,12)=32.01, N = 5). (h) Theta phase difference between IC-LFPs and LFPs recorded in CA1 pyr layer. Gray lines represent individual theta wave's phase relative to CA1 pyr theta trough (at 0/2π radians). Average phase is represented by the colored thick line. Length of the thick line represents ICPC.

The online version of this article includes the following figure supplement(s) for figure 1:

**Figure supplement 1.** Example of application of ICA.
**Figure supplement 2.** Error in phase estimation in function of theta power.
**Figure supplement 3.** Examples of synchronization using ICPC.

support the use of multichannel recordings and source separation tools to investigate interactions between theta and gamma current generators in multiple layers of the hippocampal formation.

The relative phase and coherence of these components was first compared with the theta oscillation recorded at the pyramidal layer of CA1, commonly used as the reference for temporal interactions in the hippocampus. We measured coherence and the inter-cycle phase clustering index (ICPC) which, in addition to a measure of coherence, computes the phase differences between signals in a cycle by cycle basis (see Materials and methods). All IC-LFPs exhibited prominent coherence with the LFP signal mainly in the theta range (and its first harmonic, *Figure 1g*). Similarly, the coherence at theta frequency was high between IC-LFP pairs (0.41, 0.31, 0.61, for Sch-lm, Sch-PP and lm-PP, respectively, *Figure 1g*), being larger between EC-associated generators (p<0.0001, ANOVA with degrees of freedom corrected by Greenhouse-Geisser, $F(1.091, 4.363)=89.33$). To illustrate the contribution of source separation analysis (ICA) to these results, we compared the coherence between IC-LFPs against that of LFP signals recorded at the site with the maximum contribution to each IC-LFP (i.e. *str. radiatum*, *lacunosum-moleculare* and *hilus* for Sch-IC, lm-IC and PP-IC, respectively; *Figure 1g*, dashed lines). This comparison revealed higher coherence between raw LFPs at different frequency bands, likely due to volume conduction between channels, not present in the IC-LFPs. For the theta band, the differences were most evident in the *stratum radiatum*, where theta coherence was significantly lower between IC-LFPs (*Figure 1g*). Differences were also notable in the gamma bands for all regions, which will be relevant in further analysis below. Note at this point that the extraction of highly coherent signals is perfectly compatible with ICA. This methodology finds components that are spatially distributed, and only requires small differences in their temporal co-variation (i.e. temporal jitter and/or amplitude variation). Therefore, ICA allows the separation of sources, even if there is a high coherence between them (*Makarova et al., 2011*; see Materials and methods).

The coherence results were confirmed by the ICPC analysis, demonstrating a significant coupling in the theta range with the reference LFP signal (ICPC = 0.50/0.73/0.61 for Sch-IC/lm-IC/PP-IC vs. CA1 LFP, respectively, p<0.0001, surrogate test), which also showed the characteristic phase shift across layers ($\pi/2$, $0.8\pi$ and $1.1\pi$ radians for Sch-IC, lm-IC and PP-IC, respectively; *Figure 1h*). Interestingly, the lack of coherence closer to the unit between theta oscillations in the IC-LFPs was already suggesting the coexistence of different theta current generators with certain degree of independency, rather than the artificial breakdown of a unique theta rhythm into spatially segregated components. In the latter, the coherence between the oscillations should be maximum, as they would be three components of one single wave.

## Different theta frameworks coexist in the dorsal hippocampus

To provide direct evidence of the independency between theta current generators, we next used an optogenetic approach (*Figure 2*). Using a transgenic rat cre line (LE-TG[Pvalb-iCre]2Ottc, NIDA, USA) and adeno-associated virus (AAV1-EF1a-DIO-hChR2(H134R)-eYFP-WPRE-hGH, Penn Vector Core, USA) injected in the dorsal CA3 (*Figure 2a*), we expressed the excitatory Channelrhodopsin-2 (ChR2) in parvalbumin positive (PV+) interneurons (*Figure 2b*, Materials and methods). Animals (n = 3) implanted bilaterally with optic fibers targeting dorsal CA3 and multichannel electrophysiological recordings as before (*Figure 1*) were used to test the hypothesis that hippocampal theta generators can be modulated independently by activating CA3 PV-interneurons and decreasing the Schaffer collateral output. As shown in *Figure 2c–d*, blue light illumination (460 nm) in animals freely-exploring an open field significantly and specifically decreased the power of theta in the Sch-IC and the corresponding pathway-specific low gamma oscillations. In contrast, the oscillatory activities (power and peak frequency) in lm-IC and PP-IC were preserved (*Figure 2c*). This finding was highly robust across animals (*Figure 2e*). The modulation of theta power specifically in the Sch-IC with the preservation of peak theta frequencies in the three generators, conclusively demonstrated the existence of independent theta oscillators in CA3 and EC.

We next investigated in more detail the functional interactions between the three IC-LFP theta frequencies. To reduce the error of the theta phase estimation to less than 1% of the theta cycle, we selected for further analysis only those epochs in which theta power was four times higher than delta (1–4 Hz) activity (Materials and methods and *Figure 1—figure supplement 2* for a mathematical validation of this threshold). As expected from the results in *Figure 1g and h*, theta interactions between pathways were not constant in time, but presented periods of high and low synchronization

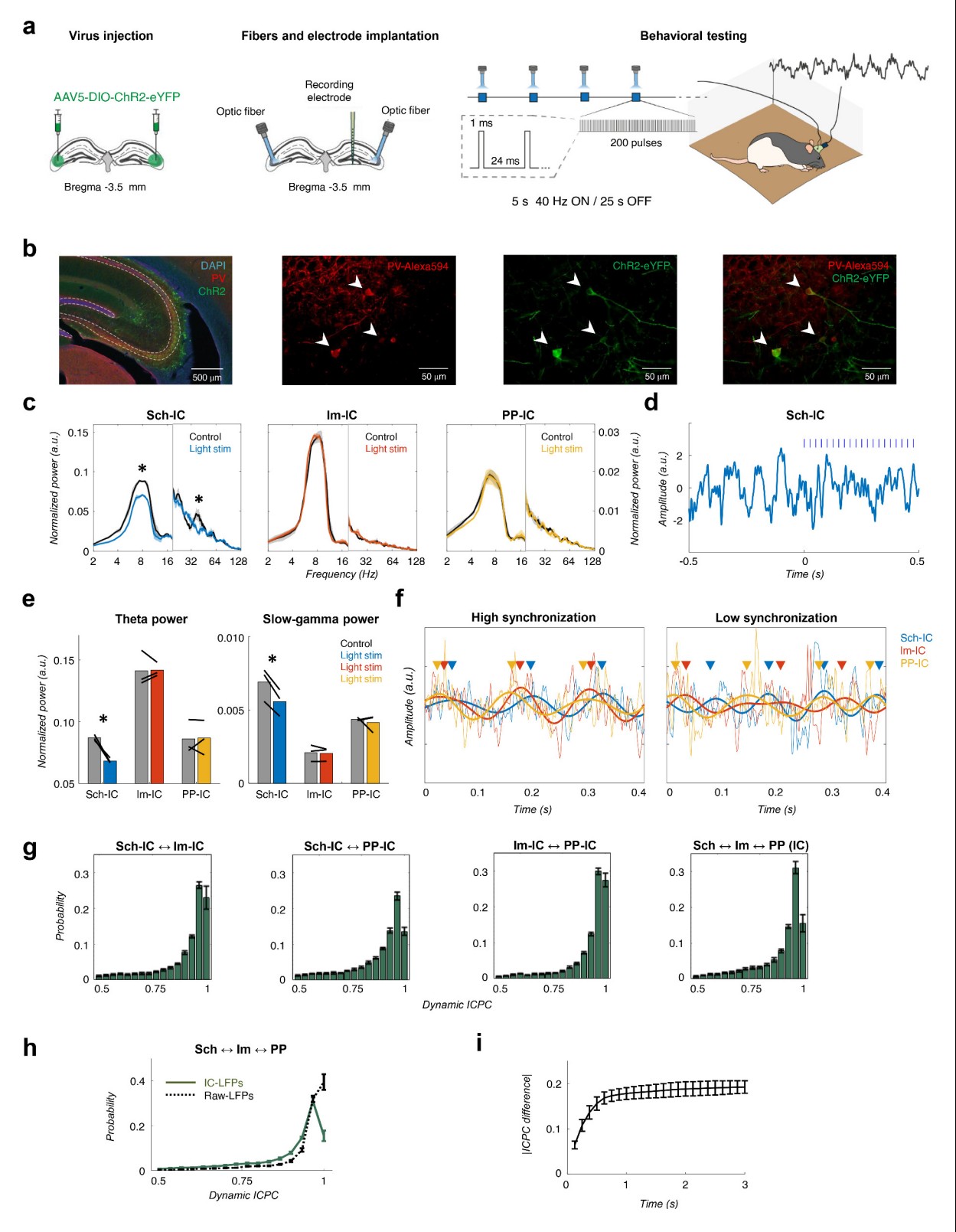

**Figure 2.** Different theta frameworks coexist in the dorsal hippocampus. (**a**) Schematic representation of virus injections (left), implantation of optic fibers and multichannel electrode (middle) and experimental setup and stimulation protocol (right). (**b**) Representative images of coronal sections confirming the specificity of ChR2 infection in PV interneurons in dorsal CA3. Left, low magnification image showing overlapped DAPI staining (blue), PV + immunostaining (red) and ChR2-eYFP expression (green). Right, higher magnification images from the CA3 region showing PV+ immunostaining

*Figure 2 continued on next page*

*Figure 2 continued*

(red), ChR2-eYFP expression (green) and their co-localization. White arrows point to double PV+ and eYFP+ interneurons. (**c**) Optogenetic manipulation of PV interneurons. Power spectra analysis (mean ± s.e.m.) of the IC-LFPs during light OFF (black traces) and light ON (colored traces) conditions. Blue light (470 nm) illumination reduced theta and slow-gamma power selectively in Sch-IC. Note different y-axis scales for low (<20 Hz) and high (>20 Hz) frequencies for visualization purposes. (**d**) Representative example of changes in theta rhythm in Sch-IC during stimulation. Blue lines represent light pulses. (**e**) Statistical comparison between theta (left) and slow-gamma power (right) in control (grey bars) and during light stimulation (color bars; *p<0.05, paired t-test, t = 7.88/7.34 for theta/gamma Sch-IC, N = 3). Black lines represent different subjects. (**f**) Traces of raw and theta-filtered IC-LFPs showing epochs with high (left) and low (right) phase locking between their rhythms. Triangles pointing to the peak of theta cycles in each IC-LFP (differentiated by colors as before) are used to highlight variability in phase differences. (**g**) Distribution of ICPC values per theta cycle between pairs of components or with the three IC-LFPs simultaneously (mean ± s.e.m). (**h**) Distribution of ICPC values between the three IC-LFPs (green lines) and between the raw LFPs. (**i**) Dynamics of ICPC changes. The y axis represents the average ICPC differences in absolute values between two cycles, and the x axis represents the time difference between their occurrences. Consecutive cycles demonstrate more similar ICPC values than those separated in time up to 0.75 s.

The online version of this article includes the following figure supplement(s) for figure 2:

**Figure supplement 1.** Representative values of ICPC in one animal along time.

(*Figure 2f*). To get insight into these states, we computed a dynamic ICPC for each theta cycle, measuring the variation of the phase relationship between theta oscillations with respect to the previous and consecutive cycles (Materials and Methods, *Figure 1—figure supplement 3*). The dynamic ICPC was measured for all pairs of IC-LFPs and for the three signals simultaneously (*Figure 2g*, *Figure 2— figure supplement 1*). The distribution of the ICPC over all animals showed a peak close to perfect phase locking, but with an important tail of low-synchronized epochs, with approximately the 20% of the cycles presenting an ICPC lower than 0.8. Again, the highest synchronization was found between PP-IC and lm-IC, in agreement with the coherence analyses (*Figure 1g*) and consistent with the likely origin of these generators in two sublayers of the same cortical regions (EC). As we did for the coherence analysis, we also computed the ICPC from the raw LFP signals recorded at *str. radiatum*, *lacunosum-moleculare* and *hilus*. This analysis showed significantly higher estimates of phase coupling based on LFPs than for IC-LFPs (averaged ICPC = 0.88/0.93 for the three IC-LFPs/raw LFPs, p<0.01, paired t-test, t = 5.35, N = 5). Perfect phase locking (ICPC = 1) was strongly reduced in IC-LFP signals, unveiling the spurious coupling measured on the raw LFPs likely due to the mixture of sources by volume conduction. Therefore, by separating the sources, ICA allowed us to clearly identify shifts in theta synchronization across hippocampal layers.

In the previous analysis, we used three consecutive theta cycles to compute the ICPC. We took this value as a trade-off between time resolution and a robust estimation of the metric. However, recent works have demonstrated that theta dynamics in the hippocampus may rapidly change between single cycles (*Dvorak et al., 2018*; *Lopes-Dos-Santos et al., 2018*; *Zhang et al., 2019*). To overcome this limitation and understand better the temporal dynamics of the theta couplings, we analyzed the variability of the ICPC across time by comparing the value of a given cycle to that of the previous ones (*Figure 2i*). These results provided a monotonously rising curve up to 0.75 s; from there on, the curve hardly increased. This indicates that the coupling strength between consecutive cycles spreads on a time scale in the order of one second, thus expecting dynamical changes in the ICPC in this time scale. Overall, this methodology allowed the characterization of the temporal synchronization between theta generators with a time resolution of one theta cycle, highlighting dynamical changes in the coupling strength between hippocampal pathways in the theta range.

## Theta-gamma CFC reflects pathway-specific interactions

The above results supported the coexistence of different temporal frames in the theta range to organize hippocampal activity. Thus, since gamma activity is nested to the theta cycle, it opened the possibility to multiple theta-gamma interactions (*Figure 3a*). For comparison, we first followed a conventional approach to the analysis of theta-gamma phase-amplitude CFC, taking as a phase reference the theta in the LFP recorded in the pyramidal layer, as is usually done (*Colgin, 2015*; *Colgin et al., 2009*; *Csicsvari et al., 1999*; *Fernández-Ruiz et al., 2017*; *Lasztóczi and Klausberger, 2016*; *Lasztóczi and Klausberger, 2014*; *Schomburg et al., 2014*; *Tort et al., 2009*; *Tort et al., 2008*), and using the modulation index (MI) introduced by *Tort et al., 2008* (*Figure 3b*). This analysis identified the typical coupling between CA1 theta and a slow gamma band of CA3

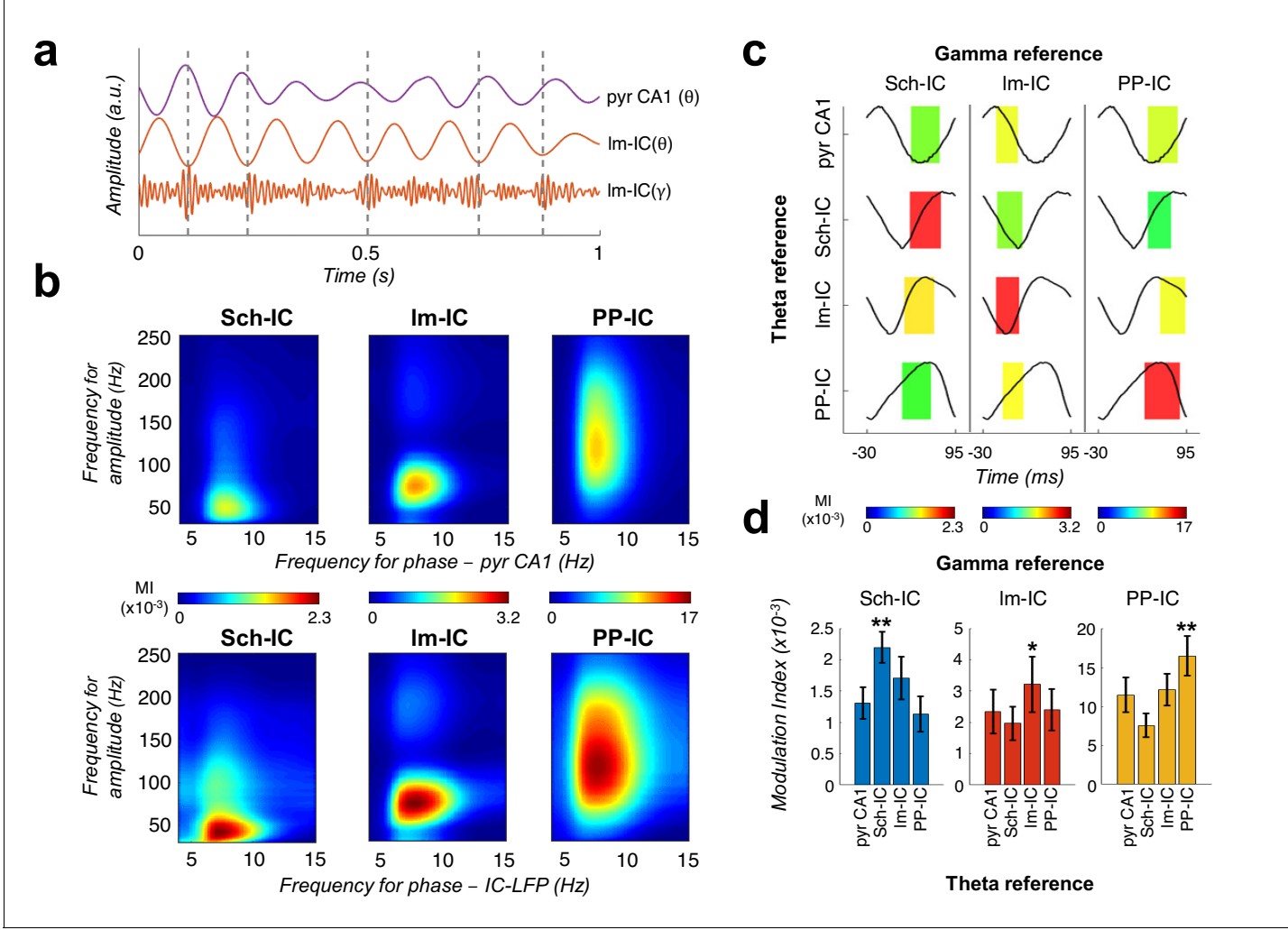

**Figure 3.** Theta-gamma coupling reflects pathway-specific interactions. (a) Representative theta- and gamma-filtered traces of lm-IC showing how the gamma envelope is phase-locked at the trough of the theta oscillation recorded in the same IC-LFP (highlighted with dashed lines), but not with the oscillation recorded in the CA1 pyramidal layer (pyr CA1). (b) Modulation strength (color coded MI) of gamma amplitude (30–250 Hz) in the IC-LFPs and the theta phase recorded in the pyr CA1 LFP (upper panels) and the pathway-specific thetas in the corresponding IC-LFPs (lower panels). (c) Interregional CFC across all gamma and theta oscillations recorded in the three IC-LFPs and pyr CA1. Each rectangle represents the MI between the theta phase and the gamma amplitude at a single specific frequency (gamma reference: slow/medium/fast gamma for Sch-IC/lm-IC/PP-IC). The location and width of the rectangles indicate the theta phase at which gamma amplitude is coupled, and the color indicates the MI (color scales at the bottom of each column). Theta waveforms (black traces) are extracted as the average of all theta cycles in the corresponding signals. The highest CFC strength (red rectangle) was always found between theta and gamma oscillations of the same pathway. (d) Maximum theta-gamma CFC corresponds to oscillations recorded in the same IC-LFP, higher than any between-pathway combination. Left/middle/right panel represents the MI (mean ± s.e.m.) between slow/medium/fast gamma recorded from Sch-IC/lm-IC/PP-IC, and the theta phase of all IC-LFPs and the LFP from pyrCA1. Significantly stronger MI values were found in all cases when theta phases were calculated from the corresponding pathway-specific generators, in contrast to pyr CA1 LFP, and when theta and gamma oscillations had the same origin (*/**p<0.05/0.01, one-way ANOVA of repeated measurements between MI with the same theta reference, followed by Bonferroni correction, F(1.182, 4.729)=36.16/F(1.133, 4.531)=8.649/F(1.555, 6.219)=35.32 for Sch-IC/lm-IC/PP-IC as theta reference, N = 5).

The online version of this article includes the following figure supplement(s) for figure 3:

**Figure supplement 1.** CFC between gamma amplitude and theta phase of each IC-LFP following the index proposed in *Canolty et al., 2006*.
**Figure supplement 2.** Effect of theta asymmetry in gamma power.

origin (Sch-IC; maximal modulation at 37.5 ± 5 Hz; *Colgin et al., 2009*; *Lasztóczi and Klausberger, 2014*; *Schomburg et al., 2014*) and a medium gamma band of EC3 origin (lm-IC; 82.5 ± 4 Hz medium gamma; *Colgin et al., 2009*; *Lasztóczi and Klausberger, 2014*; *Schomburg et al., 2014*). The analysis also revealed an additional theta-nested fast gamma band (130 ± 10 Hz,) in the mid-

molecular layer of the DG overlapping the terminal field of EC2 inputs, compatible with the previously found theta-gamma CFC in the DG (*Bragin et al., 1995*). We then computed the CFC using as references the different theta oscillations separated in the IC-LFPs. The key new finding was the systematic observation of stronger theta-gamma CFC in IC-LFPs vs. LFPs (*Figure 3b*). We further tested the robustness of these phase-amplitude CFCs by using the alternative methodology proposed in *Canolty et al., 2006* (*Figure 3—figure supplement 1*), obtaining similar results. The result was not totally unexpected, since we had found that the theta oscillation recorded in the LFP and typically used as a reference in CFC analysis was indeed a mixture of different theta generators of variable coherence (see *Figure 2d–e* above).

It has been argued that, in the case of low gamma frequencies in the hippocampus, the measured theta-gamma CFC could be a spurious effect due to the asymmetry in the theta wave (*Belluscio et al., 2012*; *Cole and Voytek, 2019*; *Cole and Voytek, 2017*) and/or theta harmonics (*Juhan et al., 2015*; *Lozano-Soldevilla et al., 2016*). This limitation, however, can be mitigated by a definition of the theta oscillation that takes into account its asymmetry instead of just applying a band-pass filter at theta frequency (*Belluscio et al., 2012*; *Cole and Voytek, 2019*; see Materials and methods). We checked the effect of theta asymmetry in our dataset with a multiple linear regression analysis, where the power at each gamma band was determined by theta power and asymmetry (*Colgin, 2016*; *Zheng et al., 2015*; *Figure 3—figure supplement 2*). We also included running speed as it has been shown to co-vary with the power and frequency of hippocampal gamma oscillations (*Ahmed and Mehta, 2012*; *Zheng et al., 2015*). We considered two factors to measure theta asymmetry: the ratio between the duration of rise and decay phases in each cycle and the ratio between the duration of the peak and the trough (*Cole and Voytek, 2018*; see Materials and methods). The analysis confirmed the influence of theta power and speed on gamma power (*Zheng et al., 2015*; *Figure 3—figure supplement 2*), and a negligible contribution of theta asymmetry. This result supports the existence of a genuine low-gamma activity band in CA1 (*Colgin et al., 2009*; *Dvorak et al., 2018*; *Lasztóczi and Klausberger, 2014*; *Schomburg et al., 2014*; *Tort et al., 2009*; *Zhang et al., 2019*) and the physiological value of its coupling with the theta oscillation.

We finally asked whether pathway specific gamma activities were preferentially coupled to the theta oscillation in their same afferent pathway, likely reflecting local computations, or in different pathways, thus reflecting inter-pathway interactions, or both. Results in *Figure 3c–d* demonstrated a dominant CFC between oscillations recorded in the same IC-LFP. Therefore, theta-gamma CFC mainly reflects pathway-specific interactions, rather than a unique carrier theta wave to which the gamma activity from different origins is multiplexed in segregated theta-gamma channels. The higher theta-gamma CFC found by using pathway-specific theta references, provided yet another indication of the coexistence and relevance of distinct temporal theta frameworks in the hippocampus.

## High theta-gamma CFC is associated to synchronization between theta frameworks

Having shown that theta generators can be modulated independently and present variable synchrony (*Figure 2*) and gamma nesting is pathway-specific (*Figure 3*), we next explored the theta and gamma features accounting for the different synchronization states. We found that theta power in all IC-LFPs correlated with the ICPC, with larger theta power associated with states of higher synchronization (*Figure 4a, b and c*). Interestingly, the frequency of the theta oscillation was constant across synchronization states in the Sch-IC, but varied in the two EC-associated generators (*Figure 4a*). Theta frequencies in lm-IC and PP-IC increased with ICPC (*Figure 4b and c*). Regarding gamma activity, broadband power did not correlate with theta synchronization (not shown), in contrast to narrowband power (slow/medium/fast gamma for Sch-IC/lm-IC/PP-IC, respectively), which correlated with the ICPC in lm-IC, but not in Sch-IC nor PP-IC (*Figure 4b*).

Because running speed also correlates with hippocampal theta power and frequency (*Vanderwolf, 1969*), we performed a multiple linear regression analysis including running speed, theta power and theta frequency as explanatory variables to predict the ICPC (*Figure 4c*). This analysis allowed us to estimate the contribution of each variable to the ICPC that cannot be accounted by any other variable in the model. We used for the analysis all theta cycles recorded while animals were exploring a familiar open-field. The MI was not included in the multiple linear regression since

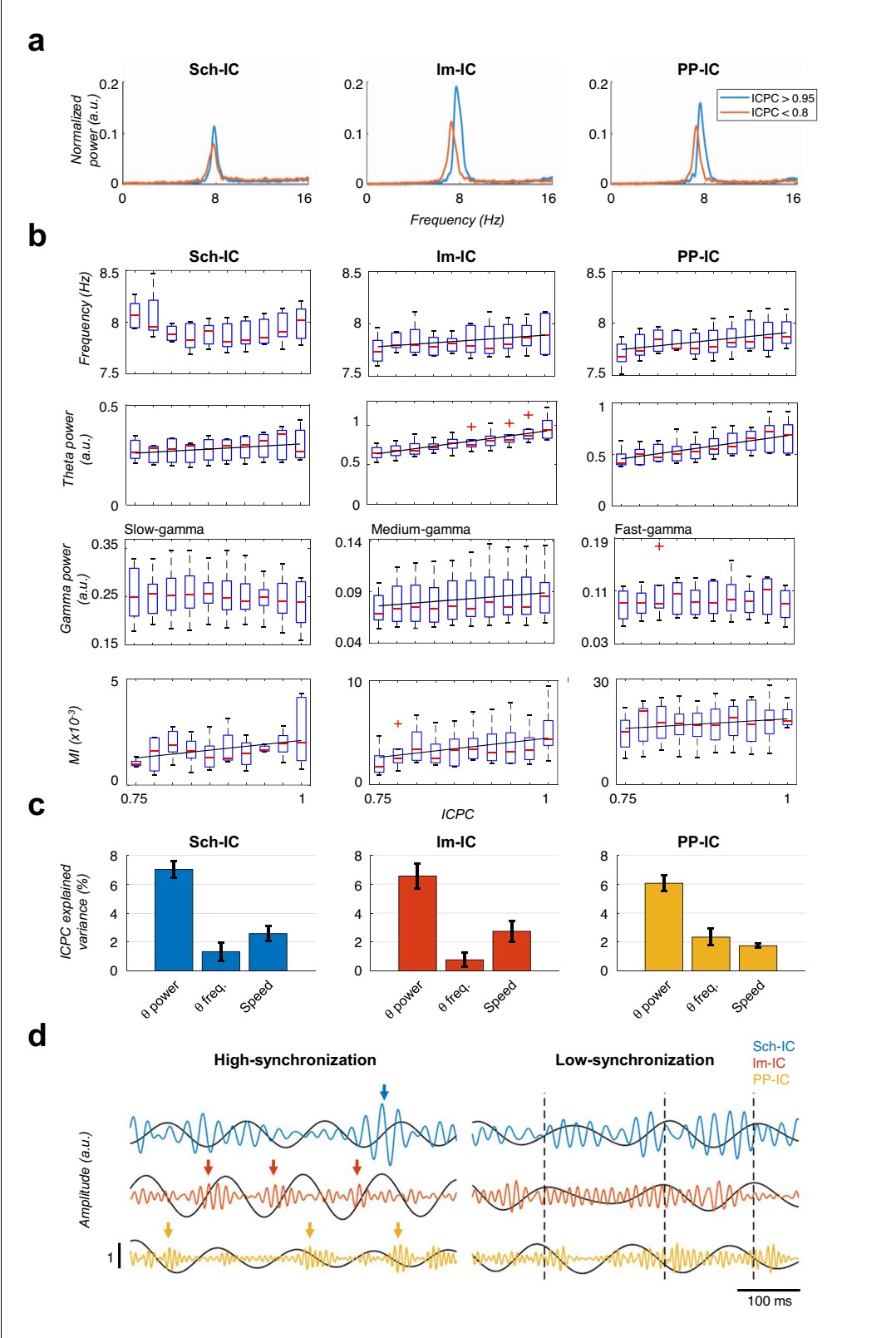

**Figure 4.** Characterization of theta oscillations as a function of their synchronization. (**a**) Power spectrum of the IC-LFPs during high (blue, ICPC >0.95) and low (red, ICPC <0.8) theta synchronization epochs. A strong increase of the theta peak can be seen in all IC-LFPs during theta synchronization, together with a right-shift of the peak frequency for lm-IC and PP-IC. (**b**) Theta frequency correlated with ICPC in lm-IC and PP-IC (black lines represent statistically significant linear correlations; R = 0.94/0.92, p<0.05, respectively, surrogate test). Theta power correlated in all IC-LFPs with the

*Figure 4 continued on next page*

*Figure 4 continued*

synchronization state (R = 0.93/0.99/0.99, p<0.05/0.0001/0.0001, respectively). Medium-gamma band was correlated with lm-IC (R = 0.96, p<0.0001), but not with Sch-IC nor PP-IC. CFC in all pathway-specific generators increased with ICPC (R = 0.71/0.82/0.77, p<0.01/0.05/0.001, respectively). Correlations were computed on the mean values of each ICPC bin. For all figures, the central mark of the box indicates the median, and the bottom and top edges of the box indicate the 25th and 75th percentiles, respectively. The whiskers extend to the most extreme data points not considered outliers, and the outliers are plotted individually with red asterisks. (c) Multiple linear regression analysis including theta power, theta frequency and speed as factors to predict the ICPC. Bars represent the variance explained by each factor that cannot be accounted for by other variables. For all cases, the contribution was considered significant (see Materials and methods). (d) Representative theta and gamma traces showing differences in CFC and theta frequency in two synchronization states, from recordings with comparable theta power. Arrows represent gamma events and dashed lines are located at the peak of the theta phase of lm-IC to facilitate the comparison of the synchronization between rhythms.

its value for individual theta cycles is not reliable. The result demonstrated the main effect of theta power on the ICPC value, with a lower, but significant, contribution of theta frequency and running speed (*Figure 4c*, p<0.05, t-test against zero between beta values of each factor, Bonferroni corrected, Materials and methods).

Finally, this analysis unveiled a striking correlation between the CFC and theta synchronization (*Figure 4b and d*). Strong theta-gamma modulation was associated with high ICPC values, while weak or nearly absent CFC was found in periods of low synchronization. Note that, as mentioned above, only cycles with high theta power activity were selected in this analysis (*Figure 4d*), so that signal's power could not affect the estimation of its phase (*Figure 1—figure supplement 2*), thus preventing the introduction of any bias in the synchronization measurement (*Figure 1—figure supplement 2*, Materials and methods). This result indicated that within-pathway CFC was associated to the synchronization between pathways.

## Gamma oscillations consistently precede theta waves

Theta and gamma oscillations reflect the extracellularly added excitatory and inhibitory synaptic and active dendritic currents of two processes occurring at different timescales (*Herreras, 2016*). We hypothesize that CFC may reflect a mechanism through which fast excitation-inhibition interactions organize the activity of principal cells in different theta frameworks found in our analysis. We then looked for an indication of directionality in the interaction between the two frequencies, and computed the cross-frequency directionality index (CFD; *Jiang et al., 2015*), based on the phase-slope index to compute the phase difference between two signals. This methodology was specially developed to estimate the directionality between signals with large differences in signal to noise ratio, as theta and gamma frequencies, demonstrating in these conditions to be more efficient than classical approaches such as Granger Causality (*Granger, 1969*; *Jiang et al., 2015*; *Nolte et al., 2010*). In CFD, an increase of the phase difference between the theta phase and the gamma amplitude with frequency gives rise to a positive slope of the phase spectrum (i.e. a positive CFD value) when the phase of the slow oscillation consistently precedes the amplitude of the fast, this is, when the time difference between a theta phase and the next burst of gamma activity is constant. The slope is negative when the amplitude of the fast oscillation consistently precedes the phase of the slow or, in our analysis, when the delay from the gamma activity to the next theta cycle is constant. As shown in *Figure 5a* for the group data, and *Figure 5—figure supplement 1* for individual animals, CFD resulted in negative values (amplitude-phase coupling, APC) for the specific gamma bands nested to the theta oscillations in the corresponding IC-LFPs. This gamma amplitude to theta phase directionality is given by the consistent anticipation of the gamma activity to the theta phase. In *Figure 5b*, there is a representative example of the gamma-to-theta directionality in PP-IC. The delay from gamma to theta is almost fixed, while in the opposite direction (theta to gamma) is highly variable. It should be noted that the CFD is not exempt of limitations and, as the CFC, the presence of harmonics and theta asymmetries may result in spurious measurements of directionality (*Lozano-Soldevilla et al., 2016*).

To validate the finding, we also computed the CFD directly in the LFP signals recorded in the different hippocampal layers. To compare with the IC-LFPs, we chose the LFP signals from the channels matching the site of maximum contribution to each IC-LFP (*Figure 5c and d*). Negative values of CFD were found in the LFPs in *str. lacunosum-moleculare* and in the DG, supporting the driving role of gamma oscillations over the phase of the theta waves. We could not find a significant

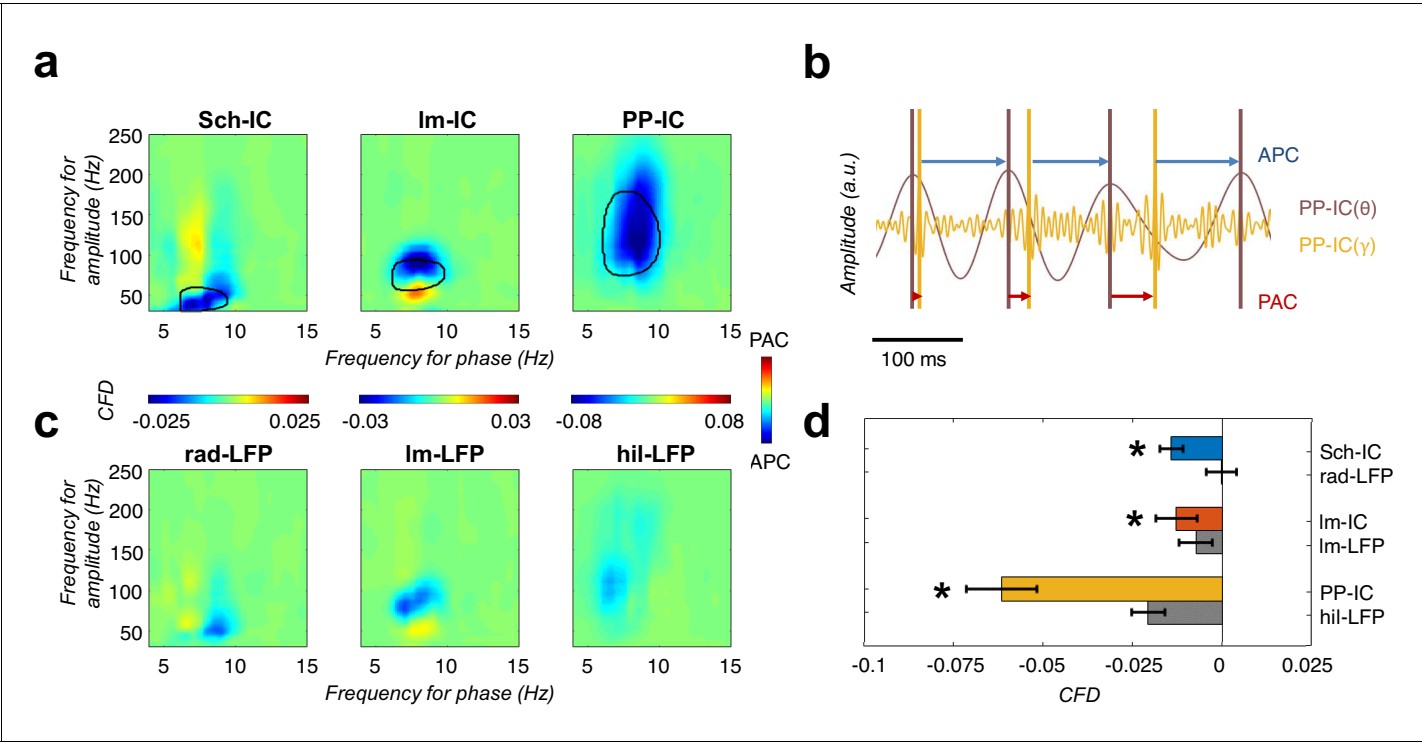

**Figure 5.** CFD analysis reveals that gamma activity modulates the phase of theta. (a) CFD analysis of the pathway-specific signals demonstrates maximum negative values (APC) for those pairs of theta-gamma oscillations with the highest CFC (encircled area). These results suggested that gamma oscillations modulate the theta phase. (b) Example of gamma-to-theta coupling in the PP-IC. The time difference from the maximum of gamma activity to the theta peak is almost fixed (APC, blue arrows), while the distance from theta to gamma varies in each cycle (phase-amplitude coupling, PAC; red arrows). (c) Same CFD analysis as in (a) using the raw LFPs from different hippocampal layers confirmed the APC directionality. (d) Comparison between CFD values using IC-LFPs and LFPs showed convergent results, with IC-LFPs outperforming the raw signals (*p<0.05, paired t-test across subjects between IC and LFP values for each IC-LFP separately, t = 3.99/2.98/4.59 for Sch-IC/lm-IC/PP-IC, N = 5).

The online version of this article includes the following figure supplement(s) for figure 5:

**Figure supplement 1.** CFD for individual animals measured along all the recording time.

directionality in the case of the *str. radiatum* LFP and, in all cases, the absolute value of the CFD was higher using IC-LFPs than LFPs (*Figure 5d*), demonstrating that source-separation tools outperform the use of raw LFPs to investigate pathway interactions. Overall, our CFD analysis suggests that the neuronal circuits supporting gamma oscillations in the hippocampus set the timing of principal cells activity in the theta range, as reflected in the phase of the recorded theta oscillations.

## Behavioral modulation of theta-gamma CFC and theta synchronization

Previous studies have shown that both CFC and inter-regional coherence, independently, correlate with learning (*Canolty et al., 2006*; *Engel et al., 2001*; *Fries, 2015*; *Fries, 2005*; *Palva et al., 2005*; *Tort et al., 2009*; *Tort et al., 2008*). Our analysis (*Figure 4*) now showed that both phenomena seem to be linked. Therefore, in our final set of experiments, we looked for behavioral evidence in support of the hypothesis that they are part of a common mechanism to flexibly integrate or segregate neuronal computations. More specifically, we hypothesized that layer-specific interactions would phase-lock theta oscillations between layers to facilitate the integration of CA3-mediated and EC-mediated information streams in CA1; for instance, in learning conditions requiring the comparison of context representations from memory (Sch-IC pathway) and from the environment (lm-IC and PP-IC pathways; *Buzsáki and Moser, 2013*; *Dudai and Morris, 2013*; *Wang and Morris, 2010*).

One such learning conditions is mismatch novelty (*Lever et al., 2006*), in which the subject is re-exposed to a previously visited context which has been modified. The 'mismatch' occurs when comparing the expected representation from memory and the found representation in the environment. To test whether ICPC and CFC increase in parallel during mismatch novelty, we trained the animals in

a task in which, after habituation to an open field (8 min session one per day during 8–10 days, *Figure 6a* control), we introduced a novel tactile stimulus in the floor of the otherwise unchanged field

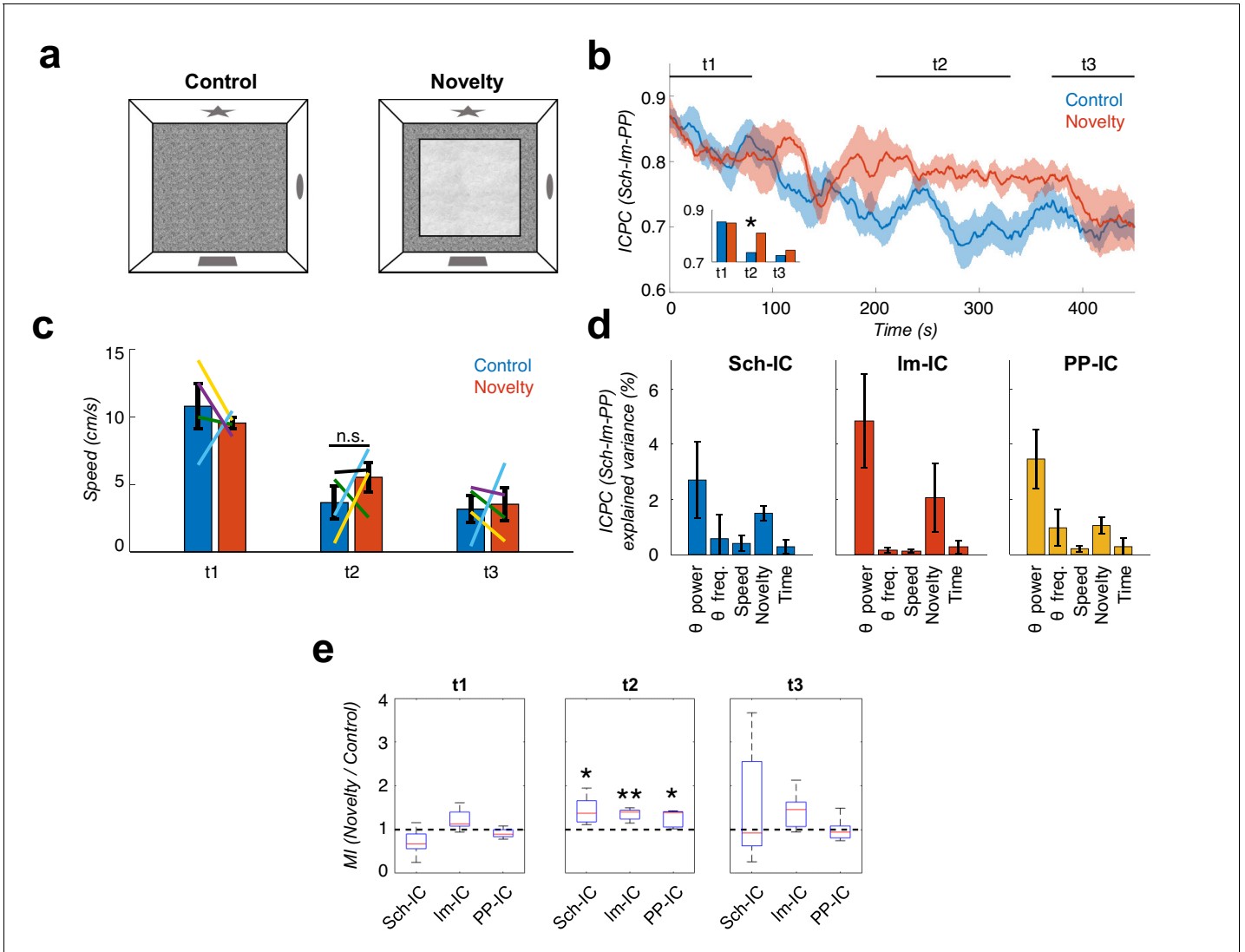

**Figure 6.** Local theta-gamma CFC and theta synchronization increase in parallel during mismatch novelty. (a) Scheme of known (left) and novel (right) open-field contexts. After the habituation period (control), the animals were exposed to a different floor (sand paper) located inside the familiar open field, providing a new tactile stimulus (novelty). (b) Time evolution of the dynamic ICPC between the three IC-LFPs (mean ± s.e.m. across all subjects) during exploration: before (blue) and after (red) the introduction of the novel tactile stimulus. Both conditions have a maximum ICPC value at the beginning of the task (t1), corresponding to the initial exploration, followed by a decay in control but not in novelty (t2, inset *p<0.05, paired t-test comparing the average ICPC in control vs. novelty for each time period separately, t = 3, N = 5). Both conditions decrease to the same ICPC level by the end of the exploration time (t3). (c) Averaged movement velocity of the animals during control and novelty. There were not significant differences between both conditions (paired t-test across subjects for each time period, N = 4). Color-lines represent the values of each subject. (d) Multiple linear regression with theta power, theta frequency, speed, session (control or novelty) and time as independent factors contributing to the ICPC between the three IC-LFPs. Bars represent the variance explained by each factor that cannot be accounted for by other variables. All contributions were significant (see Materials and methods) except for the theta frequency in Sch-IC and lm-IC. (e) CFC computed as the ratio between the averaged MI in the defined time window (t1, t2 and t3) in the novelty condition with respect to the control one (*p<0.05, paired t-test across subject, t = 2.95/5.65/2.92 for Sch-IC/lm-IC/PP-IC in t2, N = 5).

The online version of this article includes the following figure supplement(s) for figure 6:

**Figure supplement 1.** Effect of rearing in novelty task.

**Figure supplement 2.** Ratio of theta frequency in novelty sessions with respect to control sessions and for three different time windows during the task (*p<0.05, paired t-test, N = 5).

(*Figure 6a* novelty, see Materials and methods). We computed and compared theta synchrony and CFC between the novelty session and the habituation session the day before. When the animal entered the arena, the ICPC between theta oscillations was high and comparable in both conditions during the first two minutes of exploration (*Figure 6b*, t1). As the animal explored the context, synchronization remained high during novelty, but rapidly decayed in the known environment (*Figure 6b*, t2). Consistent with the notion of information transmission to update an existing memory, by the end of the exploration time both conditions decreased to the same level of theta synchronization (*Figure 6b*, t3), when the introduced tactile stimulus had lost its novelty. As a control, we tested locomotor activity comparing movement velocity between novelty and habituation sessions (*Figure 6c*), without finding differences between sessions (p>0.3, t-test, *Figure 6c*). Therefore, differences in the ICPC cannot be solely explained by changes in locomotor activity. We used a multiple linear regression analysis as before, to investigate now the independent contribution of theta power and frequency, experimental condition (control vs. novelty sessions), running speed and time in the task to the measured ICPC (*Figure 6d*). We found that theta power and the experimental condition are the main factors that contribute to the ICPC value, with other variables such us running speed and time marginally contributing.

We then computed the CFC index (MI) in the same recordings and found that it paralleled the changes in theta synchronization during the complete session in both conditions, as shown in *Figure 6b and e*. The CFC strength was higher in the three IC-LFPs during the novelty sessions associated with the higher theta synchronization, and decreased towards the end of the session in parallel with the ICPC (*Figure 6e*). Previous studies have shown that CFC in EC pathways preferentially occurs when the animal is rearing on its hind legs, an exploratory response to novelty (*Lever et al., 2006*), which is also associated with increased theta frequency (*Barth et al., 2018*). To investigate the potential contribution of rearing behavior to our findings in the mismatch novelty task, we removed from our recordings the epochs in which animals were rearing on their hind legs and then reanalysed ICPC and CFC. As shown in *Figure 6—figure supplement 1*, the increased theta synchronization between the three IC-LFPs was maintained during novelty in the absence of rearing epochs. Similarly, the MI was higher during novelty, although more variable, likely reflecting the decrease in the number of data samples after rearing removal. We concluded that the changes found in mismatch novelty cannot be solely explained by the rearing behavior. Furthermore, we measured theta frequency in the complete time series and compared it between control and novelty conditions, and found a significant decrease for Sch-IC and PP-IC theta frequency in t2 (*Figure 6—figure supplement 2*; *Wells et al., 2013*).

In a second behavioral experiment, a hippocampus-dependent delayed spatial alternation task was used in which the animal needed to remember the arm visited in the previous trial and to update the memory with the choice made in the current trial (*Ainge et al., 2007*; *Montgomery and Buzsáki, 2007*; *Wood et al., 2000*), again relying on the interaction between context representations from memory and from external sensory cues. Rats learned in an 8-shaped T-maze to alternate between the left or right arms on successive trials for water reward (*Figure 7a*) until they reached performances above 80%. In this task, the central arm is associated with memory recall, decision making and encoding of the current decision (*DeCoteau et al., 2007*; *Montgomery and Buzsáki, 2007*; *Tort et al., 2008*; *Wood et al., 2000*), while neuronal recordings in the side arm are thought to convey little information to predict behavioral outcomes in the following trial (*Pastalkova et al., 2008*; *Schomburg et al., 2014*). Using this task, previous independent studies showed a phase shift between theta oscillations recorded in CA1 and CA3 pyramidal layers (*Montgomery et al., 2009*) and increased CFC in the CA1 *radiatum* and *lacunosum-moleculare* IC-LFPs (*Schomburg et al., 2014*) associated to the central arm. Therefore, in this analysis we wanted to validate our hypothesis finding concomitant increases in theta-gamma CFC and theta ICPC in the central arm, and extend previous findings by incorporating the PP-IC into the analysis. We computed and compared theta-gamma CFC and theta synchronization in recordings obtained from the central and side arms in correct trials, selecting only those epochs were the movement velocity was comparable in both arms (*Figure 7b*). We found significantly increased CFC in the central arm for the three IC-LFPs (*Figure 7c*). Importantly, concomitant with CFC, we also found an increase in theta ICPC in the central arm during the same epochs (*Figure 7d and e*).

The results of the two behavioral tasks, thus, demonstrate that theta-gamma CFC and theta ICPC across generators are linked and preferentially occur during memory-guided exploration and

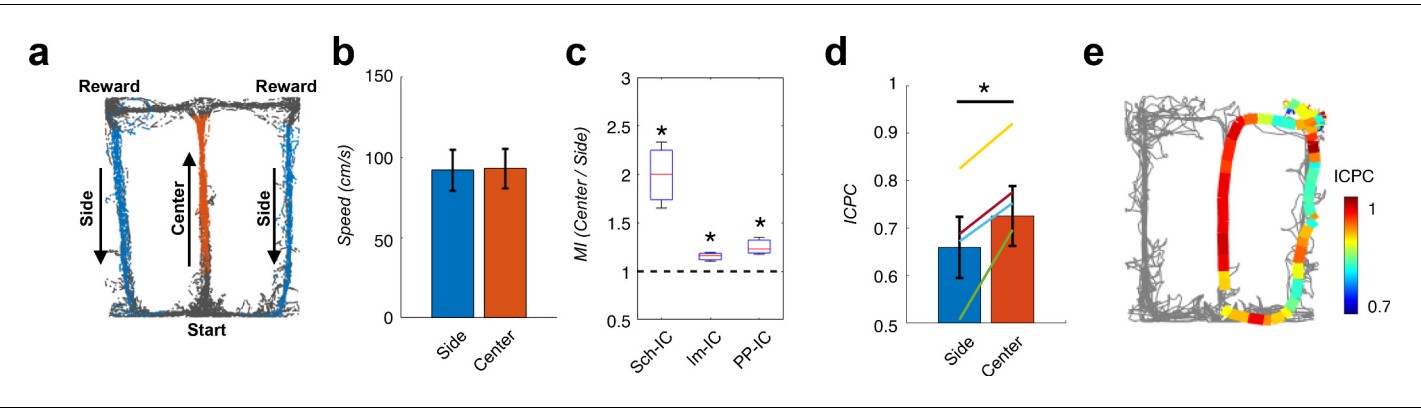

**Figure 7.** Local theta-gamma CFC and theta synchronization change in parallel and associated with decision making in a T-maze task. (a) Example of running trajectories during the T-maze task. (b) Averaged movement velocity in the center and side arms (mean ± s.e.m. across all subjects). Only trials with similar speed in both conditions were considered for the analysis. (c) Ratio between the MI at the center of the maze and that at the sides (*p<0.05, paired t-test across subjects, averaging all selected trials for each animal, t = 3.61/4.03/3.54 for Sch-IC/Im-IC/PP-IC, N = 4). (d) ICPC values in the center and side arms showing the synchronization between the three IC-LFPs (*p<0.05, group level paired t-test, t = 4.79, N = 4). Differences within-subject (color lines) were done with paired t-test using all correct trials (yellow line: p<0.0001, N = 114 trials; purple: p<0.05, N = 63; green: p<0.001, N = 151; blue: p=0.07, N = 69). (e) Representative example of ICPC values for different locations in one trial.

mismatch novelty detection, two conditions in which internally generated memory representations need to be integrated with the incoming sensory information about external cues. They support the idea that different theta-gamma frameworks may flexibly coordinate information transmission in the hippocampus.

## Discussion

Overall, our results provide functional evidence supporting independent theta oscillations in the hippocampus whose coordination can be seen as a mechanism to channel information between hippocampal layers. Synchronized theta states may bind distributed computations, while less synchronized theta states may secure relatively independent processing in local circuits (segregation). What might be the mechanism coordinating theta frameworks? We show that theta phase locking across hippocampal layers is associated with stronger CFC. Furthermore, directionality analysis demonstrates that band- and pathway-specific gamma activity consistently precede theta waves, possibly contributing to the synchronization of theta oscillations across layers. We thus hypothesize that the CFC reflects a mechanism operated by local excitation-inhibition interactions to coordinate neuronal computations in separated theta frameworks. In a network with multiple connected nodes, theta-phase locking between specific nodes would further contribute to the directionality of the information flow, habilitating targets between which communication is permitted or prevented in defined time windows. We have provided evidence supporting this hypothesis by showing that CFC and the coordination between the theta current generators recorded in the hippocampus increase in the mnemonic process.

### Independent theta frameworks

In this work, we have separated the LFP sources contributed by different synaptic pathways using spatial discrimination techniques based on independent component analysis (*Benito et al., 2014*; *Fernández-Ruiz and Herreras, 2013*; *Herreras, 2016*; *Makarov et al., 2010*; *Makarova et al., 2011*). This processing step allowed us to work with a more reliable representation of the local electrophysiological dynamics, as compared to raw LFPs or CSDs (*Martín-Vázquez et al., 2013*). The main drawback of LFPs is the multisource origin of the signals, a blend of dipolar (or quadrupolar) field potentials. While the CSD of multisource raw LFPs avoids the problem of volume conduction, it does not separate the co-activating current sources in the recorded region, hence the CSD of pathways targeting the same cells (e.g. CA1 pyramidal cells) overlap and add/subtract, cancelling each other. In these conditions, the time course of the CSD is a composite one (as it is that of native

LFPs) and cannot be unambiguously assigned to any of the co-activating sources. Therefore, source separation techniques are necessary to obtain the correct time course of each individual synaptic contribution.

Extensive previous research has demonstrated the existence of multiple theta rhythms and current generators in the hippocampus and EC (*Buzsáki, 2002*). While septal activity is required for theta rhythmicity, and lesions targeting the medial septum eliminate theta oscillation in both structures, intrinsic hippocampal activity from CA3 and extrinsic EC inputs do also contribute to the recorded theta oscillations (*Buzsáki, 2002*). Surgical removal of the EC unveils a theta oscillation that depends on the integrity of CA3 and is highly coherent across hippocampal layers (*Bragin et al., 1995*). In the presence of an intact EC, however, the coherence between theta signals in the *stratum radiatum* and *lacunosum-moleculare* is reduced (*Kocsis et al., 1999*), consistent with an input competition between CA3 and EC3. Now, using optogenetic tools targeted to CA3 PV+ interneurons (*Figure 2*), we provide new results that conclusively support the coexistence of independent theta oscillations in the hippocampus, by showing the specific modulation of the CA3-associated Sch-IC *vs.* the EC-associated theta generators (lm-IC and PP-IC). Furthermore, variations in theta power and frequency in each generator occurred dynamically and independently (*Figure 4*). Nevertheless, phase shifts between the identified theta frameworks served to coordinate them in pairs or triads, in a sub-second time-scale (*Figure 2* and *Figure 2—figure supplement 1*). In turn, we speculate, these hippocampal theta coupling states would be associated with distinct brain-wide network states. In support of this view, we found selective behavioral/cognitive functions associated with different states of between-framework theta synchronization (*Figures 6* and *7*). Finally, the dynamic change in theta frequency observed in individual pathway-specific LFPs (*Figure 4* and *Figure 6—figure supplement 2*) also argues against the view of the identified theta frameworks as monolithic oscillations driven by external pacemakers, and rather suggests the cooperation of weakly-coupled local oscillators and global rhythm generators for the fine-tuning of theta oscillations. Thus, theta activity in the hippocampus is neither unique, nor monolithic.

Overall, taking brain oscillations as rhythmic changes in neuronal excitability that can define sequential information packages (*Canolty and Knight, 2010*; *Fries, 2005*), the dynamic variation in theta synchrony found in the hippocampus likely reflects multiple theta-coordinated time-frames, with phase differences between oscillations having a large impact on the timing of neuronal firing in the respective layers. Synchronization of theta frameworks would, in turn, coordinate, though not necessarily synchronize (*Mizuseki and Buzsaki, 2014*), firing sequences in consecutive hippocampal stations. The processing streams thus generated could transmit independent information, e.g. driven by memory retrieval or external environmental cues, or the result of integrating/comparing both information sources, depending on the cognitive needs.

## Theta-gamma interactions

Interactions between the phase of the theta oscillation and the amplitude of the gamma activity have been extensively documented and proposed as an effective mechanism to integrate activity across different spatial and temporal scales (*Bragin et al., 1995*; *Bruns and Eckhorn, 2004*; *Buzsáki and Draguhn, 2004*; *Canolty et al., 2006*; *Canolty and Knight, 2010*; *Colgin, 2015*; *Colgin et al., 2009*; *Engel et al., 2001*; *Fell and Axmacher, 2011*; *Lakatos et al., 2008*; *Lakatos et al., 2005*; *Lisman and Idiart, 1995*; *Lisman and Jensen, 2013*; *Mormann et al., 2005*; *Palva et al., 2005*; *Saleh et al., 2010*; *Soltesz and Deschênes, 1993*; *Tort et al., 2009*; *Tort et al., 2008*; *Zheng et al., 2016*). Our analysis demonstrates that phase-amplitude CFC between theta and gamma oscillations in the hippocampus is selective for theta-frameworks (*Figure 3*). Interactions between theta and gamma were higher when IC-LFP theta oscillations were used as the temporal reference for the corresponding layer-specific gamma activities, instead of a single LFP recording, as commonly done. This observation, together with previous and important evidence demonstrating that firing of principal cells in CA3 and EC3 is phase-locked to downstream theta-nested gamma oscillations recorded in the CA1 *stratum radiatum* and *lacunosum-moleculare*, respectively (*Colgin et al., 2009*; *Lasztóczi and Klausberger, 2014*; *Schomburg et al., 2014*), suggests that local layer-specific circuits interact with upstream afferent pathways to organize hippocampal cell assemblies in multiple theta-gamma frameworks.

The significant positive correlation between CFC strength and theta synchronization found in our study (*Figure 4*) further suggests that layer-specific CFC might reflect the mechanism for theta

framework coordination. Co-modulation of gamma amplitude and theta phase can be the result of a theta-driven process increasing gamma activity, a gamma-driven modulation of theta phase, or due to the presence of a common external drive for both components, fast and slow, simultaneously. The CFD index has been previously shown to reveal both fast-to-slow and slow-to-fast frequency interactions in modelled data and real electrophysiological recordings (*Helfrich et al., 2019*; *Helfrich et al., 2018*; *Jiang et al., 2015*; *Zheng et al., 2017*). We applied it here, for the first time to hippocampal IC-LFPs, and unveiled, in contrast to a generalized assumption in the field, a predominant gamma-to-theta interaction (*Figure 5*). This directionality was confirmed directly on the LFP signals (*Figure 5*), although the contribution of a third input controlling simultaneously both rhythms cannot be fully discarded. We do not take this result as an indication of theta oscillations in the hippocampus being generated by gamma activity. On the contrary, we suggest that gamma activities, reflecting the interplay of inhibitory-excitatory networks (*Cardin et al., 2009*; *Neymotin et al., 2011*; *Orbán et al., 2006*; *Rotstein et al., 2005*; *Stark et al., 2013*; *Tort et al., 2007*), impose phase shifts on the on-going theta oscillations in their corresponding layers. Therefore, local gamma-generating circuits, driven by afferents from their respective upstream layers, might not be activated at a particular theta phase, but rather be coordinating principal cells activity and setting the phase of the local theta oscillation.

While dissecting the precise circuit mechanisms supporting the above gamma-to-theta interaction is out of the scope of the present work, several possibilities exist. Computational works have demonstrated that theta-gamma CFC emerges from the interactions between functionally distinct interneuron populations interconnected in a network of principal cells receiving an external theta rhythm generator, such as the septal input (*Neymotin et al., 2011*; *Orbán et al., 2006*; *Rotstein et al., 2005*; *Tort et al., 2007*). Subsets of interneurons can phase-lock to different hippocampal rhythms (*Klausberger et al., 2003*; *Klausberger and Somogyi, 2008*) and, interestingly, recent findings have shown in the CA1 region that some interneurons can specifically phase-lock to slow-gamma and others to medium-gamma, supporting the idea that different classes of interneurons drive slow and medium gamma oscillations (*Colgin, 2015*; *Fernández-Ruiz et al., 2017*; *Lasztóczi and Klausberger, 2014*). Thus, an appealing mechanism for the gamma-modulation of theta phase would be the control of different interneuron classes by pathway-specific inputs, which would entrain specific gamma networks modulating principal cell excitability and firing in response to on-going theta inputs, advancing or delaying theta phases. In support of this hypothesis, recent analyses of theta-gamma associations on a theta cycle-by-cycle basis have demonstrated a significantly higher spike-field phase synchrony for interneurons than pyramidal cells in the theta band (*Zhang et al., 2019*). Finally, spiking resonance in principal cells may contribute to this mechanism too, since optogenetic activation of basket interneurons (PV-cells) in the hippocampus and neocortex pace pyramidal cell firing in the theta range, by virtue of postinhibitory rebound of $I_h$ activity (*Stark et al., 2013*). In that experiment, theta-band firing of excitatory neurons required rhythmic activation of inhibitory basket cells, as white noise activation effectively modulated their activity but did not entrained pyramidal theta-band firing (*Stark et al., 2013*). Feed-forward activation of interneurons from upstream layers or an external rhythmic input (i.e. cholinergic or GABAergic inputs form the septum) are thus required for resonance amplification. Intrinsic cellular properties and network mechanisms may thus interact to support gamma-dependent coordination of theta phases across hippocampal layers.

The above interpretation would also explain phase-phase coupling between CA1 theta and CA1 slow- and medium-gamma (*Belluscio et al., 2012*), as the consequence of theta phase driven by pathway-specific gamma activity entrained by upstream inputs in CA3 and EC3, respectively. Recent studies, however, highlighted the importance of frequency harmonics and waveform asymmetry when measuring phase-phase coupling (*Scheffer-Teixeira and Tort, 2016*) and also amplitude-phase CFC (*Cole and Voytek, 2017*). Waveform asymmetry in oscillatory activity introduces spectral content that cannot be defined solely by sinusoidal components (*Amzica and Steriade, 1998*) and, therefore, may result in spurious CFC and CFD (*Juhan et al., 2015*; *Cole and Voytek, 2017*; *Kramer et al., 2008*; *Lozano-Soldevilla et al., 2016*; *Scheffer-Teixeira and Tort, 2016*). Several approaches have been developed to overcome these limitations (*Cole and Voytek, 2019*; *Kramer et al., 2008*), improving the estimation of the theta phase and minimizing the effect of sharp edges. We applied these methods in our analysis (Materials and methods). The specific waveform of an oscillation should not be seen as a problem but as a source of physiological information when appropriately considered (*Cole and Voytek, 2017*).

## Parallel processing, segregation and integration

The proposed scenario provides a mechanism to coordinate distributed computations organized in theta waves by synchronizing theta oscillations through theta-gamma CFC. We reasoned that layer-specific interactions would phase-lock theta oscillations between layers when the integration of CA3- and EC-associated information streams is required (*Buzsáki and Moser, 2013*; *Dudai and Morris, 2013*). We selected two well-known behavioral tasks to test this hypothesis. We first used a mismatch novelty task (*Lever et al., 2006*) in which the memory representation of the context, involving the CA3-associated pathway, is compared against the novel (mismatch) sensory input, conveyed by the EC-associated pathways (to CA1 and DG). Our hypothesis predicted that, during the novelty condition, a concomitant increase in CFC and theta synchronization in the three theta-gamma frameworks should occur, something that we found experimentally (*Figure 6*). This result could not be explained solely by the animal's speed, which was indistinguishable in our experiments between known and novelty conditions, nor by rearing behavior, sometimes associated with novelty exploration (*Barth et al., 2018*). A link between CFC and theta synchrony was also found in the central arm of the 8-shaped T-maze after correcting for running speed (*Figure 7*), in the location where the interaction between context representations from memory and from external sensory cues take place for decision making and encoding (*DeCoteau et al., 2007*; *Montgomery and Buzsáki, 2007*; *Tort et al., 2008*; *Wood et al., 2000*). These results bring together previous independent findings showing phase shifts between theta oscillations recorded in CA1 and CA3 pyramidal layers (*Montgomery et al., 2009*) and increased CFC in the CA1 *radiatum* and *lacunosum-moleculare* IC-LFPs (*Schomburg et al., 2014*) associated to the central arm.

A recent study using an uncharted novelty test showed increased theta-gamma CFC exclusively in EC pathways, but not in the CA3 pathway (*Barth et al., 2018*). Importantly, in contrast to mismatch novelty, uncharted novelty involves the exposure to a previously unvisited context and therefore it lacks a memory representation. Thus, in the absence of a memory representation, only EC-pathways conveying information about the environmental cues demonstrate enhanced theta-gamma coupling, lending support to our hypothesis. Finally, we found that theta frequency decreased in the mismatch novelty condition in the Sch-IC and PP-IC (*Figure 6—figure supplement 2*; *Wells et al., 2013*), while it was reported to increase in the EC-pathways in uncharted novelty (*Barth et al., 2018*), suggesting that theta frequency modulation was required to couple the three theta frameworks during mismatch novelty.

Important recent studies have investigated theta oscillations in a cycle-by-cycle manner (*Dvorak et al., 2018*; *Lopes-Dos-Santos et al., 2018*; *Zhang et al., 2019*), demonstrating highly dynamic changes in spectral components and theta-gamma interactions associated to different behaviors. These studies support the notion that individual theta cycles represent flexible temporal units to transiently organize CA1 computations. We found that layer-specific theta oscillations coexisting in the hippocampus couple and decouple dynamically, and we propose that it reflects a mechanism to integrate or segregate computations, respectively. This possibility is fundamentally different from previous ones based on the segregation of computations in the phase of the theta wave (*Colgin et al., 2009*; *Lisman and Idiart, 1995*; *Lisman and Jensen, 2013*), or in single theta cycles as indicated above (*Dvorak et al., 2018*; *Lopes-Dos-Santos et al., 2018*; *Zhang et al., 2019*), in that those were based on the rapid alternation of computational modes between phases or cycles, respectively, but always of a unique theta framework. In contrast, our new proposal contemplates parallel processing in cell assemblies receiving information from different theta frameworks. A decrease in the coherence between the theta oscillations would decouple the processing streams, segregating the underlying cognitive processes (i.e. retrieval from encoding). An increase in the coherence would rather couple them, facilitating the integration in CA1 neurons and downstream regions of both information streams (i.e. when stored and ongoing contextual information need to be compared). Interestingly, however, the two models complement each other, since computations in each theta framework would likely vary in a cycle-by-cycle manner, representing an even more versatile coding framework.

## Concluding remarks

Interactions between slow and fast brain oscillations have been measured in multiple brain regions during perception, attention, learning and memory formation (*Buzsáki and Draguhn, 2004*;

*Engel et al., 2001*; *Lisman and Jensen, 2013*). Despite its ubiquitous presence in fundamental cognitive processes, its function is largely unknown. Our results provide a mechanism for parallel processing in the hippocampus based on the coexistence of multiple theta frameworks that support both, segregated or integrated computations, depending on their synchronization level. Important questions remain to be answered. How theta synchronization in the hippocampus relates to hippocampal-neocortical interactions (*Siapas et al., 2005*; *Sirota et al., 2008*) known to be favoured at theta and beta frequencies (*Igarashi et al., 2014*; *Moreno et al., 2016*) and modulated by synaptic plasticity in the hippocampus (*Álvarez-Salvado et al., 2014*; *Canals et al., 2009*)? The conditions triggering the coordination between theta-gamma frameworks are not well understood, but given that theta-gamma uncoupling seems to represent an early electrophysiological signature of hippocampal network dysfunction in Alzheimer's disease (*Goutagny et al., 2013*; *Iaccarino et al., 2016*; *Palop and Mucke, 2009*; *Verret et al., 2012*) as well as for schizophrenia and other psychiatric disorders (*Olypher et al., 2006*; *Phillips and Silverstein, 2003*; *Uhlhaas and Singer, 2006*), further and detailed mechanistic investigations are granted.

## Materials and methods

All animal experiments were approved by the Animal Care and Use Committee of the Instituto de Neurociencias de Alicante, Alicante, Spain, and comply with the Spanish (law 32/2007) and European regulations (EU directive 86/609, EU decree 2001–486, and EU recommendation 2007/526/EC).

### Animals and surgery

Five male Long-Evans rats, with a weight of 250–300 g. were trained in different behavioral tasks, with a multichannel electrode recording the electrophysiological activity in the hippocampus (data are available at http://dx.doi.org/10.20350/digitalCSIC/12537). The sample size was selected based on previous reports with analysis of hippocampal theta and/or gamma in a T-Maze task (*Montgomery and Buzsáki, 2007*; *Schomburg et al., 2014*; *Tort et al., 2008*). All of them were implanted with a 32 channels silicon probe (Neuronexus Technologies, Michigan, USA) connected in turn to a jumper consisting of two corresponding connectors joined by 5 cm of flexible cable. An Ag/AgCl wire (World Precision Instruments, Florida, USA) electrode was placed in contact with the skin on the sides of the surgery area, and used as ground. The data were acquired at 5 kHz, with an analog high-pass filter at 0.5 Hz. After digitalization, we initially low-pass filtered them at 300 Hz, removed the net noise with Notch filters at 50 Hz and 100 Hz and down-sampled the signals at 2.5 kHz. We adjusted the final position of both electrodes using as a reference the typical evoked potentials at the dentate gyrus (*Andersen et al., 1966*), so that a maximal population spike in the dentate gyrus was recorded.

After the surgery, the rats were left for at least 10 days until they recovered completely. During the first 72 hr, they were injected subcutaneously with analgesic twice per day (Buprenorphine, dose 2–5 µg/kg, RB Pharmaceutical Ltd., Berkshire, UK). During 1 week, they had as well antibiotic dissolved in the water (Enrofloxacin, dose 10 mg/kg, Syva, León, Spain). The behavioral tasks were not started until the animals showed no signs of discomfort with the manipulation of the implants.

### Data acquisition

All subjects were trained before the surgery following the next protocol. The first three days consisted on a habituation process with two 10 min sessions per day in an open field, with freedom of movement. The environment was a methacrylate sandbox of $50 \times 50$ cm, opened at the top and with three visual cues in three of the walls. After that, they carried out two new sessions per day for 8 days, first repeating the habituation and then performing a modified T-maze task, that has been described previously (*Wood et al., 2000*). It consisted in several tracks in 8-like shape (132/102/80 cm long/wide/high with track wide 8 cm, *Figure 6e*). The starting point was located at the beginning of the central rail (*Figure 6e*, start) and the rat was forced to run across that arm (*Figure 6e*, center), blocking other pathways with black panels. At the end of the track, it must choose one of the two directions of the T-junction and a small drop of water was delivered at the corner (*Figure 6e*, reward) in successfully trials. Each repetition is considered successful if the rat chooses the opposite direction with respect to the previous trial, finding always a reward at the corner after the T-junction. Then, another panel located after the water prevented the rat from retracing its route, forcing it to

go to the starting point across the corresponding side arm (*Figure 6e*, side), for a new trial. Each session had a duration of 20 min with around 30 trials, and all the subjects reached a performance greater than 80% in the last session. Only correct trials were considered for further analysis.

After the surgery and recovery, we repeated the same protocol for 8 days. For further electrophysiological analysis, we considered only those sessions were the subjects kept a high level of performance (80%) without any interference. There were in total between 2 and 5 sessions of 4 subjects. During the 9th day, we carried out a 'novelty' test. The rats were exposed to a novelty by introducing them in a 'novelty chamber' located inside the familiar open field; such chamber was a transparent methacrylate box with a square base 35 cm wide, and 40 cm high, opened at the top, with sand paper on the floor to provide a noticeable tactile stimulus. After this time, the novelty chamber was removed, and the animals were left another 10 min in the open field, considering this session as the control condition for the analysis.

Except for the results in *Figure 6*, all the analyses were carried out during the control session (last session), with freedom of movement in a well-known environment.

## Optogenetic experiments

Four male Long-Evans transgenic rats, expressing Cre recombinase under the rat parvalbumin promoter (LE-TG[Pvalb-iCre]2Ottc, NIDA, USA), were bred in our facilities, housed in pairs with food and water available ad libitum and maintained on a 12/12 hr light-dark cycle.

## Virus injection and surgeries

For the surgery, rats were anesthetized with isoflurane (4.5% induction, 1–2% maintenance, in 0.8 l/min $O_2$) and locally anesthetized by subcutaneous injection of bupivacaine (0.2 ml). All rats weighted 300–330 gr at the time of the first surgery. The Cre-dependent viral vector AAV1-EF1a-DIO-hChR2 (H134R)-eYFP-WPRE-hGH (Penn Vector Core) was bilaterally injected in dorsal CA3 (AP −3.5 mm, LM ± 3.6 mm from bregma and DV –2.8 mm from the brain surface) using a Hamilton syringe attached to an infusion pump (1 µl per hemisphere at 1 µl/min). Thus, ChR2 is specifically expressed in PV+ cells (PV-ChR2).

Two weeks after the virus injection, rats underwent a second surgery for two fiber-optic cannulas and one recording electrode implantation. First, five screws were attached to the skull to strengthen the fixation of the implant. As in the previous group of rats, a 32 channels silicon probe connected to a jumper (Neuronexus Technologies, Michigan, USA) was placed in the left hippocampus covering dorsal CA1 and DG. Reference wires were attached to one of the screws. The coordinates for the electrode implantation were AP −3.5 mm, LM ± 2.5 mm from bregma and DV –3.0 mm from the brain surface, although its final position was adjusted based on the electrophysiological potentials evoked by stimulating the perforant pathway. Then, a fiber-optic cannula (200 µm diameter, 0.66 NA, 10 mm length; Doric Lenses, Quebec, Canada) was placed in the dorsal CA3 of both hemispheres with an angle of 20° in the coronal plane at the coordinates AP −3.5 mm, LM ± 5.2 mm from bregma and DV –3.2 mm from the brain surface (*Figure 2a*). Once the electrode and the two fibers were positioned, the stimulation electrode was removed and several layers of dental cement (Super-Bond or Palacos) were applied to ensure enough fixation of all the components. The post-operative care was the same as in the previous group of rats (see above). One of the subjects was excluded from the experiment after the surgery and prior to any analysis due to the reduced quality of its electrophysiological recordings.

## Optogenetic manipulation in behaving animals and data acquisition

All behavioral procedures were conducted during the dark cycle. After the complete recovery following the implantation surgery and before starting the experiments, we handled the animals during 5 days in order to habituate them to the experimenter as well as to the manipulation of the implant (connection of the headstage and the fiber-optic patch cords).

Rats performed an 8 min daily session during 5 consecutive days in an already known open field (50 × 50×40 cm box of black methacrylate) with bedding covering the floor. Animals were allowed to freely explore the arena in each session while receiving ON/OFF periods of light stimulation. The light for excitation of ChR2 was delivered at 50 mW/mm² by a blue LED source (Prizmatix, Canada)

at a wavelength of 460 nm. Stimulation protocol consisted on 5 s 40 Hz trains with 1 ms light pulses each 30 s during the entire session.

For the bilateral stimulation, we used a branching fiber-optic patch cord (500 µm diameter, 0.63 NA; Doric Lenses) connected to a rotatory joint (Prizmatix) which in turn connects to the LED source by a fiber-optic patch cord (1 mm diameter, 0.63 NA; Doric Lenses). The power density of the delivered light was measured prior to each session using a powermeter (Thorlabs) to ensure the same power density in all sessions (50 mW/mm$^2$).

Light pulses were triggered by a stimulus generator (STG2004, Multichannel Systems, Reutlingen, Germany) controlled by MC_Stimulus software (Multichannel Systems). Electrophysiological data were recorded at 5 kHz sampling rate with an open-source acquisition system (Open Ephys) and synchronized with light stimulation and video recording by using an I/O board (Open Ephys).

## Inmunohystochemical analysis

After the performance of the experiments, the rats were perfused intracardially with PFA 4%. Brains were kept in post-fixation for 3 hr at RT and then stored in PBS at 4°C o/n. Then, brains were cut in 50µm-slices to corroborate the viral infection as well as the correct position of the recording electrode and the fiber-optic cannulas. Slices were incubated with monoclonal PV antibody developed in mouse (1:2000, Swant, Switzerland) and afterwards with an anti-mouse secondary antibody developed in goat (1:500, Alexa Fluor 594 dye, Life Technologies, USA). After completion of histological treatments, brain sections were imaged using a fluorescence microscope (DM4000B, Leica) coupled to a Neurolucida software (MicroBrightField, Inc) and images were processed with Image J software.

## Video recording and tracking

The animals were monitored during all the tasks, and their behavior was recorded using a standard camera located at the top of the room. Using those videos, the location of the subjects was tracked with the software tracker (physlets; https://www.physlets.org/tracker/), taking their centroid as the refence point. The synchronization of the video and the electrophysiological recordings was made triggering a red LED and matching the temporal mark that it left in the recordings with the first frame with light.

## Current source density analysis of LFPs

The first approach to achieve the information of the sources contributing to the LFPs was the use of CSD analysis (*Freeman and Nicholson, 1975*; *Herreras, 1990*; *Holsheimer, 1987*; *Mitzdorf, 1985*). It measures the transmembrane currents, providing a spatiotemporal distribution of the local sinks and sources (inward and outward currents, respectively). Contrary to the LFPs, these currents represent spatially localized phenomena, increasing the spatial resolution.

The membrane currents can be achieved following the Laplace equation and using the measured field potentials and the conductivity of the medium. As the hippocampus is a layered structure, the one-dimensional approach in the direction parallel to the recording electrode was used:

$$CSD_m(t) = -\frac{\sigma}{h^2}(u_{m-1}(t) - 2u_m(t) + u_{m+1}(t)), \tag{1}$$

where $u_m(t)$ is the LFP recorded at the m-th site, $h$ is the distance between channels and $\sigma$ is the conductivity of the extracellular space.

We assumed the whole structure as an isotropic and homogeneous medium. Though hippocampal strata present different resistivities, they do not affect much to the temporal dynamics of specific locations (*Herreras, 1990*; *Holsheimer, 1987*; *López-Aguado et al., 2001*). Therefore, the distance and conductivity are constants, and they merely act as a scale factor. In this work, the distance between contacts was $h$ = 100 µm and we assumed a constant conductivity $\sigma$ = 350 $\Omega^{-1}$cm$^{-1}$ (*López-Aguado et al., 2001*).

Though the CSD presents higher spatial resolution than the LFPs, it does not discriminate contributions from different pathways. Multiple membrane currents with different origins may overlap spatiotemporally, and local currents are also affected by the activity in nearby domains (*Herreras, 2016*; *Korovaichuk et al., 2010*; *Martín-Vázquez et al., 2013*). To overcome these limitations, we applied an independent component analysis (ICA).

## Independent Component Analysis of LFPs

Methods such as CSD analysis make it possible to isolate the local transmembrane currents by eliminating propagated field potentials (see above). Nevertheless, different pathways are contributing to these currents, and their activities may overlap spatiotemporally. To disentangle the specific sources that generate the LFPs, we applied an ICA.

The effectiveness of this approach has been well studied and established in the hippocampus (*Fernández-Ruiz et al., 2012b*; *Herreras et al., 2015*; *Korovaichuk et al., 2010*; *Makarov et al., 2010*; *Makarova et al., 2011*; *Schomburg et al., 2014*). It aims to solve the problem of separating N statistically independent sources that have been mixed in M output channels. To do that, it performs a blind separation of patterns, because the different distributions of the sources are unknown. Moreover, it assumes spatial immobility of the sources or, in other terms, a fixed location of the axon terminals. The contribution of their synaptic currents to the LFP conforms the different independent components (ICs) or generators to unravel.

Each recorded time-series $u_m(t)$ is modeled as the sum of N neuronal sources multiplied by a constant factor:

$$u_m(t) = \sum_{n=1}^{N} V_{mn}s_n(t), \;\; m = 1, 2, \ldots, M, \tag{2}$$

where $V_{mn}$ is the mixing matrix with the voltage loadings of N LFP generators on M electrodes and $s_n(t)$ is the time-series associated to the n-th LFP generator.

As the number of ICs with significant variance is usually low (4–7 out of 32) (*Benito et al., 2014*; *Korovaichuk et al., 2010*), we applied a dimension reduction of the loading matrix by prior use of principal component analysis, keeping 99% of the original LFP variance. For each structure and electrographic state, the number of optimal components is determined by stepwise increase of the number of principal components until the new ICs are only noise (*Makarov et al., 2010*). Since noisy components contribute negligible variance (in absence of artefacts in the signal) we always choose this number plus two. There are several algorithms to compute the mixing matrix that transform LFP data into ICs, nevertheless, all of them share a common theory framework. In this work, we have used the information-maximization approach RUNICA (*Bell and Sejnowski, 1995*), implemented in the matlab toolbox 'ICAofLFPs' (http://www.mat.ucm.es/~vmakarov/downloads.php). For comparison purposes, the kernel density ICA algorithm KDICA (*Chen, 2006*) was also computed, obtaining similar results.

By definition, the ICA may extract as many generators as the number of LFP signals. To correctly identify the presynaptic specificity of an IC, several conditions must be taken into account. First, each IC contributes differently to the total variance (power) of the LFP. Only those with a significant contribution (>1% in this work) were considered for further analysis. Second, the anatomic structure of each generator is fixed in each subject (*Castellanos and Makarov, 2006*; *Korovaichuk et al., 2010*; *Makarov et al., 2010*). In other words, the spatial profile of each ICs must be stable along the time. This was assessed by applying the ICA in different short-term epochs (*Korovaichuk et al., 2010*). Only those components present in all conditions with a stable spatial loading may represent true current generators. Moreover, a certain degree of similarity is expected between subjects and a specific pathway should have a comparable profile for different animals. Third, the synaptic specificity of each generator was determined by stimulating their respective excitatory pathways with subthreshold evoked activity (*Makarova et al., 2011*). Fourth, not every synaptic input leaves a footprint in the LFP. The geometry of the region and the distribution of axons and dendrites determine the real contribution of each pathway to the field potential (*Buzsáki et al., 2012*; *Herreras, 2016*; *Herreras et al., 2015*). This requires specific realistic models to test the multiple origins of the measured currents.

Applying ICA in our data recordings and considering all the conditions mentioned above, we were able to extract three common and stable generators in all subjects (*Figure 1* and *Figure 1—figure supplement 1*). They correspond to pathway-specific inputs to the hippocampus. Two of them were in CA1: one in *str. radiatum*, which corresponded to the synaptic terminals of Schaffer collaterals from CA3 to the pyramidal cells in CA1 (*Benito et al., 2014*; *Fernández-Ruiz et al., 2012a*; *Korovaichuk et al., 2010*; *Makarova et al., 2011*; *Martín-Vázquez et al., 2016*; *Schomburg et al., 2014*) (Sch-IC); the other component had a current sink in *str. lacunosum-*

*moleculare* (lm-IC), where are located the inputs from EC3 to the pyramidal cells in CA1 (*Benito et al., 2014*; *Martín-Vázquez et al., 2016*; *Schomburg et al., 2014*). A third component was identified in the DG, which corresponded to the axons projected from the EC layer II (EC2) to the dendrites of the granular cells through the perforant-pathway (*Benito et al., 2014*; *Korovaichuk et al., 2010*; *Makarova et al., 2011*) (PP-IC). Note that the active synaptic domain of PP-IC was in the molecular layer of the DG, but its field potential was dominant in the hilar region (*Figure 1*). This was generated by the volume conduction of the cell membranes into the molecular layer. The field potentials of common currents, above and below the hilus, are overlapped in this region, thus increasing their electric field (*Benito et al., 2014*; *Herreras, 2016*; *Herreras et al., 2015*).

The extracted ICs represent the current sources of specific pathways to the hippocampus. Therefore, the temporal dynamics and rhythms of each generator reflects the activity generated in different nearby regions (CA3, EC3 and EC2 for Sch-IC, lm-IC and PP-IC, respectively). One limitation of this approach is that it cannot separate distinct temporal patterns within the same origin (i.e. theta and gamma oscillations). This is the consequence of two main effects. First, the same neuron could fire in multiple modes (*Vinogradova, 2001*). Moreover, the currents generated by synaptic terminals from the same region are fully overlapped in the space and their combination made up a single generator. It should be noted at this point that ICA requires independence in space and time. Therefore, two spatially separated sources with exactly the same temporal dynamics would converge in a single component. Nevertheless, small differences in the signals' co-variation (i.e. temporal jitter and/or imperfect amplitude co-variation) would allow a correct separation of the two sources, even if there is a high coherence between them (*Makarova et al., 2011*).

Another consideration is that the strongest generators could introduce contamination in other components (*Korovaichuk et al., 2010*; *Schomburg et al., 2014*). To ensure that the huge theta power is not affecting the discrimination of ICs, we separated the LFP in slow and fast rhythms by filtering the raw LFPs at 30 Hz (low-pass at <30 Hz and a high-pass at >30 Hz, respectively). The ICA was applied to each filtered data-set separately. We compared the resultant ICs with those using the unfiltered data, confirming that the same generators were found in all conditions with quite similar time-series (*Figure 1—figure supplement 1*).

ICA does not ensure the correct polarity and amplitude of each generator. However, as the ICA algorithm is invertible, the LFPs generated by each component can be retrieved separately. The CSD can be applied to these reconstructed signals, obtaining the sinks and sources of each specific pathway. Such partial signals do have the correct polarity and amplitude (*Korovaichuk et al., 2010*; *Martín-Vázquez et al., 2013*).

## Preprocessing and power analysis of time series

After computing the ICA algorithm, the dataset corresponding to each subject was composed by three time-series. These signals were downsampled at 625 Hz to improve the speed of computational analysis. They were also normalized, imposing to each dataset an averaged mean value of zero and a standard deviation of one to each signal separately. This way, we increase the similarities inter-subject and facilitate their comparison.

Power spectra were estimated using the multitaper method (*Thomson, 1982*). For power analysis at specific frequency bands we used an approach based on filtering and Hilbert transform (*Jackson et al., 2006*; *Ólafsdóttir et al., 2017*). First, the signal is bandpass filtered with a FIR filter between the frequencies of interest. Then, we computed the Hilbert transform and the instantaneous power was estimated as the squared complex modulus of the signal at each time point. The mean value was obtained as the averaged power in a certain time window. We defined the following frequency bands which are used along the text (unless otherwise indicated): delta (1–4 Hz), theta (6–10 Hz), slow gamma (30–60 Hz), medium gamma (60–100 Hz) and fast gamma (100–150 Hz).

The linear interaction between IC-LFPs at each specific frequency was assessed using a coherence analysis. It measures the ratio between the cross power spectral density and their individual power spectral densities and was computed using the mscohere.m function in Matlab. The statistical significance was determined by a surrogate analysis (1000 surrogates in this work). With this methodology, the temporal relationship between signals was broken by randomly displacing one signal respect to the other. Then, the coherence surrogated results at each frequency were approximated to a

Gaussian distribution and the significance threshold was the value for which the previous cumulative distribution was 0.95 (p=0.05).

To evaluate the distribution of the gamma activity along the phase of theta, the signals were first filtered at the frequencies of interest (gamma and theta) and the amplitude and phase were extracted using the Hilbert transform. Then, for each theta cycle, the envelope of the gamma activity was divided into N equidistant bins; an average along all cycles was then taken. Similar to the coherence analysis, the statistical significance was assessed by a surrogate analysis (1000 surrogates), randomly shifting the gamma signal with respect to the theta phase. The surrogate distribution was estimated by averaging the results of all simulations.

## Detection of theta rhythm

From the whole recordings, only those epochs with a real theta rhythm in all components where considered for further analysis, that is, with high power at that band. Moreover, as oscillations with low amplitude could result in a less accurate estimation of their phase, we selected only those cycles with a minimum value of theta power to avoid this issue. To find such threshold we have modeled theta rhythm data as the combination of theta oscillations ($X_\theta(t)$) and pink noise ($X^n(t)$) to simulate the neural noise of the recordings:

$$X_\theta^n(t) = X^n(t) + X_\theta(t) \tag{3}$$

The noise was computed using the function *pinknoise* from Matlab, while $X_\theta(t)$ was composed by $d$ segments or cycles defined as:

$$S_i^k(t_i) = A(\sin(2\pi f_i t_i + 1.5\pi)) \tag{4}$$

Where $k = 1, 2, \ldots, d$ and $f_i t_i$ where selected in order to that cycle had a duration of $T_i \in [0.1, 0.145]$ seconds, randomly chosen from a normal distribution with mean 0.125 and standard deviation 0.02.

Knowing each $S_i^k(t_i)$, we could estimate perfectly the phase of the theta rhythm, that is this was the ground truth. Briefly, for each simulation we changed the relative power between both components by varying the $A$ parameter and we estimated the phase of the theta oscillation (see below). Doing so, we were able to measure the error between our estimation and the ground truth as a function of the relative theta power and then find the value that minimizes that error. A detailed description of these steps follows.

For each dataset, we bandpass filtered each signal at delta (1-4 Hz) and theta (6-10 Hz) frequencies. Then, we computed the Hilbert transform and the instantaneous power was estimated as the squared complex modulus of the signal at each time point. The relative theta power was obtained as the ratio between the averaged theta by the delta power (*Ólafsdóttir et al., 2017*; *Jackson et al., 2006*). The phase of the oscillation was estimated for the theta filtered signal through Hilbert, being zero and π radians the values corresponding to the trough and the peak of the cycle, respectively. Finally, the error was measured as the averaged distance (in milliseconds) between each trough of the real phase and the estimated one. Additionally, we computed the minimum error as that obtained when $X^n(t)$ is set to zero. This value corresponds to the noise introduced by the method used to estimate the phase and does not depend on the power. The relationship between the ratio and the error is shown in *Figure 1—figure supplement 3*.

Using the simulated data, we considered that the theta power was not influencing the estimation of the theta phase when the error due to the amplitude (i.e. not considering the one introduced by the filtering) was lower than 1 ms. This corresponded to a ratio of theta power 3.78 times (we took four for simplicity) higher than delta (*Figure 1—figure supplement 3*). In the real recordings we expect not only neural noise but also activity at the delta band. Thus, the theoretical threshold obtained by this procedure represent a conservative measurement. For those cases with high delta activity, the threshold would be more restrictive as the ratio decreases, but the theta power would be always high enough to guarantee a correct estimation of the phase.

To detect the theta rhythm in all the ICs recorded, we used a sliding window of one theta cycle (approximately 125 ms) and selected only those epochs where the ratio between their theta power by delta (computed as described above) was higher than 4. Moreover, analysis taken thresholds of 6 and 8 were also done for comparison purposes, showing no significant differences.

## Optogenetic modulation of pathway-specific theta activity

In the Cre-expressing transgenic subjects we analyzed whether the light stimulation of CA3 PV-inter-neurons had an effect exclusively in the Schaffer collateral theta output. We compared time windows of two seconds immediately after the stimulation with the first two seconds of the stimulus. This period was chosen to minimize the influence of different locomotor activities between windows. For all trials, only those with theta oscillation (see above) were further considered. Then, we computed the power spectrum of each IC-LFP using a multitaper approach (*Thomson, 1982*).

## Inter-cycle phase clustering

To estimate the relative theta phase between ICs, we used a modified inter-trial phase clustering approach (*Cohen, 2014*) to account for differences between cycles instead of trials (ICPC). In this methodology, each trial is defined as a vector with modulus one and the angle corresponding to the phase of the oscillation measured at a specific time point ($\varphi_t$). Then, the ICPC is computed as the modulus of the averaged vector among all trials:

$$ICPC = \left| \frac{1}{N} \sum_{t=1}^{N} e^{i\varphi_t} \right| \tag{5}$$

If the distribution of angles is uniform along the polar axis, then the ICPC value is zero. On the contrary, values near one indicate a preferred phase in the distribution, being one when all trials have the same phase.

For the analysis in *Figure 1h*, the phases of each signal were extracted following *Cole and Voytek, 2019* to have a better characterization of the shape of the theta rhythm. The LFP measured in the pyramidal layer of CA1 (pyr. CA1) was considered as reference, where its trough and peak coincide with 0 and $\pi$ radians, respectively. We calculated the ICPC for the different components separately, where each trial was the phase of the IC measured at each trough of pyr. CA1. Thus, the number of trials corresponded to the number of theta cycles in pyr. CA1 and the angle and value of the ICPC can be interpreted as the phase difference and the stability of the IC with respect to pyr. CA1, respectively.

The statistical significance was assessed by a surrogate analysis (1000 surrogates in this work), randomly shifting the phase of the IC and keeping pyr. CA1 constant. For each simulated dataset, the ICPC was computed fitting all the surrogated results into a normal distribution. Then, the p-value associated to the ICPC of the IC was obtained as one minus the previous normal cumulative distribution evaluated at the ICPC value.

## Cycle-by-cycle synchronization using ICPC

Using the ICPC approach, the degree of coherence between theta rhythms was estimated for each cycle separately. Doing this process, a dynamic measurement of synchronization can be done identifying time epochs of high and low coherence. Considering two ICs, one acting as the reference, the ICPC value of each theta cycle was computed using only three cycles, which corresponded to the relative phase at that cycle and the previous and consecutive ones (*Figure 2—figure supplement 1*). If the waves were highly coherent, then their phase would be similar along theta cycles, resulting in an ICPC value close to 1; for those with different phases, the ICPC would be lower. In this work, we considered the theta oscillation in lm-IC as the reference to compute the ICPC, as it had the highest amplitude at that frequency. Moreover, the ICPC between pairs of signals (i.e. lm-IC vs. Sch-IC and lm-IC vs. PP-IC) was averaged as an approximation of the global synchronization of the network at each time epoch.

The instantaneous ICPC can be compared to other metrics as frequency or power, analyzing the correlation between synchronized state and the features of the signals. To compute each correlation, the data were classified into 10 groups as a function of the ICPC, with ten equidistant bins from 0.75 to 1. We chose these values as they contain at least the 90% of the cycles in all subjects and 80 cycles per bin (around 10 s). Then, the averaged value of the different metrics was computed for each group, analyzing the correlation between the means and the ICPC value. To identify reliably relationships, we compared if the resultant correlation value ($\rho$) was higher than the obtained by a surrogate analysis. Each simulated dataset (100 surrogates in this work) was built by randomly

shifting the IC components, breaking any temporal relationship between them. Then, the correlation between the ICPC and other features of interest were computed, fitting the results to a gaussian distribution. We considered that a correlation was significant if its value was higher than the 95th percentile of the surrogate distribution.

## Multiple linear regression

The joint analysis of several features as predictors of the ICPC was done using a multiple linear regression. Firstly, we ranked all the values associated to each feature (function tiedrank.m in matlab) to minimize the effect of outlayers in the dataset. Then, a single model is fitted using each predictor multiplied by a beta factor:

$$ICPC = \beta_0 + \beta_1 power + \beta_2 frequency + \beta_3 speed + \ldots + \varepsilon \tag{6}$$

Where $\beta_0$ is a constant value and $\varepsilon$ are the residuals.

The contribution of one specific predictor to the total explained variance of the model is estimated by fitting a reduced multiple linear regression without that predictor and computing the difference between variances in the full model minus the reduced one.

To estimate if each variable is significantly contributing to the ICPC across subjects, we followed *Montgomery and Buzsáki, 2007* and tested their associated beta values for a statistical difference from zero (t-test, Bonferroni corrected).

## Cross-Frequency coupling

Interactions between the phase of a low frequency oscillation and the amplitude of a faster one were measured using an approach based on the modulation index (MI) (*Canolty et al., 2006*; *Tort et al., 2008*). The original method computed the phases and amplitudes through filtering and Hilbert transform. Nevertheless, theta rhythms in the hippocampus are non-sinusoidal oscillations and filtering the signal results in errors at estimating the waveform shape and could introduce spurious coupling (*Cole and Voytek, 2017*; *Kramer et al., 2008*). A new methodology has been proposed to overcome this issue and estimate the instantaneous phase of this kind of oscillations (*Cole and Voytek, 2019*). Briefly, we used a combination of a narrowband filter to detect the zero-crossing points (which correspond to the ascendant and descendant slope of the oscillation, or the phases π/2 and 3π/3, respectively) and a broadband filter to find the trough and the peak (phases 0 and π, respectively). This way, the phase of the signal does not vary monotonically (as in the case of a sinusoidal one) but could follow fast changes in the cycle as an abrupt ascendant slope and a soft descent. We named this signal as $x_{\theta\varphi}(t)$.

The amplitude of the faster oscillation is computed as the envelope of the signal filtered at the specific frequency which we want to analyze. First, we used a filter centered at that frequency and with a bandwidth at least two times the frequency of the phase signal where the coupling is expected to be maximum (*Juhan et al., 2015*). Considering that the main rhythm in the hippocampus is around 8 Hz, the bandwidth should have at least 16 Hz (we used 20 Hz in this work). Then, the envelope is computed using the Hilbert transform. The resultant signal is labelled as $x_{\gamma A}(t)$.

To compute the MI, we divide each cycle of $x_{\theta\varphi}(t)$ into *N* bins. To avoid the issues introduced in this method due to the previously mentioned non-uniform theta oscillation (*van Driel et al., 2015*; *Cole and Voytek, 2017*), these bins were not of the same size, but were equalized along the phase of the theta cycle. Instead of using *N* bins, we divided each cycle into 4 segments which correspond to the epochs between the peak, the trough and both ascendant and descendant pendants (*Cole and Voytek, 2019*), and then divided that segment into *N*/4 bins of the same size. Therefore, *N*/4 bins were used from the trough to the middle of the ascendant phase, *N*/4 bins from this point to the peak and so on. After that, we computed the mean amplitude of $x_{\gamma A}(t)$ at each bin, calling $<x_{\gamma A}>_\varphi(j)$ the amplitude at the phase bin *j*. From them, we can calculate the entropy *H*, defined by:

$$H = -\sum_{j=1}^{N} p_j \log p_j \tag{7}$$

where *N* was set to 20, and $p_j$ is given by

$$p_j = \frac{<x_{yA}>\phi(j)}{\sum\limits_{j=1}^{N}<x_{yA}>\phi(j)} \tag{8}$$

The value of MI is defined as the entropy $H$ normalized by the maximal entropy ($H_{max}$), given by the uniform distribution $p_j = 1/N$ (i.e. $H_{max} = \log N$):

$$MI = \frac{H_{max} - H}{H_{max}} \tag{9}$$

A value of MI near 0 indicates lack of phase-to-amplitude modulation, while larger MI values reflect higher coupling between both signals.

The values obtained using the MI given by Equation 9 were compared to those provided by the phase-amplitude coupling index proposed by Canolty and colleagues (mean vector length; *Canolty et al., 2006*). The latter is estimated as the degree of asymmetry of the probability density function of the gamma amplitude across the phase of theta. Both methods yielded similar results.

The statistical significance has been assessed following the steps proposed by Canolty and colleagues (*Canolty et al., 2006*), by a surrogate analysis (n = 100 surrogates) in which each surrogate is built by cutting the phase signal at a random point and exchanging the resultant segments. This breaks the temporal relationship between both time-series minimizing the distortion of their dynamics (*Juhan et al., 2015*). The MI estimated among surrogates represents the coupling due to the oscillatory nature of the signals but not by a real temporal relationship. Therefore, the MI values of all surrogates are approximated to a gaussian distribution, whose 95th percentile is considered as a significance threshold.

## Cross-Frequency directionality

The MI is a measurement of the degree of interaction between the phase and the amplitude of two frequencies, but it has no information regarding the directionality in this coupling. On the one hand, the theta phase would modulate the amplitude at gamma frequencies while, on the other hand, the gamma activity could be leading the phase. To identify leader and follower in this interaction, we have used the CFD index (*Jiang et al., 2015*). The main idea is that, supposing that the phase component precedes the amplitude with a fixed time delay of $k$ ms, the distance from the peak of the phase to the next peak of the amplitude should be $k$ ms (i.e. there is an increase of gamma activity $k$ ms after the peak of theta phase). Thus, as not all phase cycles have the same duration, the distance from the peak of the gamma amplitude to the next peak of theta may vary. Nevertheless, in the case of gamma activity preceding theta, the result would be the opposite. The peak of gamma activity should appear $k$ ms before the peak of the phase, while the timing from phase to amplitude would be different for each cycle. Note that the CFD is an estimation of the temporal relations between signals, but not a measurement of causality per se as, for example, a third source could be interacting with both signals.

The CFD identifies this relationship using the phase-slope index (PSI; *Nolte et al., 2008*), a measurement of directionality between time series. Briefly, if the oscillation of one signal $x(t)$ at a certain frequency is driving a second one $y(t)$ with a time delay, then the phase difference between them at that specific delay will change consistently with the frequency of the signals. The slope of the phase is obtained in function of the frequency, and its sign will indicate who is the driver. If the slope is positive (higher the frequency, higher the phase difference between signals) then $x(t)$ leads $y(t)$ in time, while negative values would indicate that $y(t)$ precedes $x(t)$. The CFD is a variant of the PSI, where one signal is the theta component, and the other the envelope of the gamma activity.

Calling $x(t)$ to the original signal, $x_{yA}^v(t)$ to the power envelope of the signal at a $v$ gamma frequency, and being $X$ and $X_{yA}^v$ their Fourier transform, respectively, the CFD is defined as the PSI by:

$$\psi(v,f_j) = \mathrm{Im}\left(\sum_{f_j-\frac{\beta}{2}}^{f_j+\frac{\beta}{2}} C^*(v,f_j) C(v,(f_j+f))\right) \tag{10}$$

where

$$C(v,f_j) = \frac{\sum\limits_{s=1}^{S} X^s \left(X_{\gamma A}^{v,s}\right)^*}{\sqrt{\sum\limits_{s=1}^{S}|X^s|^2 \sum\limits_{s=1}^{S}\left|X_{\gamma A}^{v,s}\right|^2}} \qquad (11)$$

is the complex coherency, Im is the imaginary part, '\*' denotes the complex conjugate, $f_j$ is the theta frequency under study, $S$ is the number of segments in which the signal has been divided and β is the bandwidth for which the phase slope is measured, and it has been fixed at 2 Hz, 4 times the resolution ($f = 0.5$ Hz).

This methodology has been proposed specifically for those frameworks with phase-amplitude coupling as more classical approaches like Granger Causality (*Granger, 1969*) have some unavoidable limitations due to the use of filters (*Barnett and Seth, 2011*) and the differences of signal-to-noise ratio in both components (*Nolte et al., 2010*), which provoke that they cannot identify the correct directionality in CFC models (*Jiang et al., 2015*).

To provide statistical significance, a new surrogate test (n = 100 surrogates) has been developed following the same steps than in the MI analysis. Note that in this case, two thresholds are imposed, considering both tails of the gaussian distribution (i.e. positive and negative values). In this work, we have used the Matlab toolbox HERMES (*Niso et al., 2013*) and the implementation of PSI in Matlab code (http://doc.ml.tu-berlin.de/causality/). All the Matlab code developed to compute cross-frequency analysis is freely available at https://canalslab.com/. To emphasize the directionality in the region of higher CFC, the MI comodulogram has been redefined as a mask, with values from 0 to 1 (minimum and maximum value MI). Applying this mask to the CFD comodulogram, areas without phase-amplitude coupling are attenuated, while the main cluster remains constant.

## Acknowledgements

We thank Begoña Fernández for excellent technical assistance, Laura Pérez-Cervera for her input during data pre-processing with ICA and Raul Vicente for his advice in CFC. SC and DM were supported by the Spanish Agency of Research (AEI) under Grant Nos. BFU2015-64380-C2-1-R and −2 R, respectively, co-financed by the European Regional Development Fund (ERDF). SC was further supported by AEI and ERDF under Grant No. PGC2018-101055-B-I00, the European Union Horizon 2020 research and innovation programme under Grant Agreement No. 668863 (SyBil-AA) and acknowledges financial support from the Spanish State Research Agency, through the Severo Ochoa Program for Centres of Excellence in R and D (SEV- 2017–0723). CRM and EP acknowledge support from MINECO trough project Nos. TEC2016-80063-C3-3-R and −2 R, respectively. CRM also acknowledges financial support from the Spanish State Research Agency, through the María de Maeztu Program for Units of Excellence in R and D (MDM-2017–0711). OH was supported by MINECO under Grant No. SAF2016-80100-R. VJL was supported by a predoctoral fellowship La Caixa-Severo Ochoa from Obra Social La Caixa.

## Additional information

### Funding

| Funder | Grant reference number | Author |
|---|---|---|
| European Regional Development Fund | BFU2015-64380-C2-1-R | Santiago Canals |
| European Regional Development Fund | BFU2015-64380-C2-2-R | David Moratal |
| European Regional Development Fund | PGC2018-101055-B-I00 | Santiago Canals |
| Horizon 2020 Framework Programme | 668863 (SyBil-AA) | Santiago Canals |
| Agencia Estatal de Investigación | SEV- 2017-0723 | Santiago Canals |

| | | |
|---|---|---|
| Ministerio de Economía y Competitividad | TEC2016-80063-C3-3-R | Claudio R Mirasso |
| Ministerio de Economía y Competitividad | TEC2016-80063-C3-2-R | Ernesto Pereda |
| Agencia Estatal de Investiga-ción | MDM-2017-0711 | Claudio R Mirasso |
| Ministerio de Economía y Competitividad | SAF2016-80100-R | Oscar Herreras |

The funders had no role in study design, data collection and interpretation, or the decision to submit the work for publication.

## Author contributions
Víctor J López-Madrona, Formal analysis, Methodology, Writing - original draft, Writing - review and editing; Elena Pérez-Montoyo, Formal analysis; Efrén Álvarez-Salvado, Resources; David Moratal, Conceptualization; Oscar Herreras, Conceptualization, Software, Investigation, Methodology; Ernesto Pereda, Resources, Software, Validation, Methodology; Claudio R Mirasso, Conceptualization, Supervision, Investigation, Visualization, Methodology, Writing - original draft, Writing - review and editing; Santiago Canals, Conceptualization, Resources, Supervision, Validation, Investigation, Visualization, Methodology, Writing - original draft, Writing - review and editing

## Author ORCIDs
Víctor J López-Madrona https://orcid.org/0000-0001-8234-7160
Oscar Herreras http://orcid.org/0000-0002-8210-3710
Claudio R Mirasso http://orcid.org/0000-0003-2980-7038
Santiago Canals https://orcid.org/0000-0003-2175-8139

## Ethics
Animal experimentation: All animal experiments were approved by the Animal Care and Use Committee of the Instituto de Neurociencias de Alicante, Alicante, Spain, and comply with the Spanish (law 32/2007) and European regulations (EU directive 86/609, EU decree 2001-486, and EU recommendation 2007/526/EC).

## Decision letter and Author response
Decision letter https://doi.org/10.7554/eLife.57313.sa1
Author response https://doi.org/10.7554/eLife.57313.sa2

# Additional files

## Supplementary files
• Transparent reporting form

## Data availability
All datasets are available at: https://doi.org/10.20350/digitalCSIC/12537.

The following dataset was generated:

| Author(s) | Year | Dataset title | Dataset URL | Database and Identifier |
|---|---|---|---|---|
| Gamoneda CS, Salvado AE | 2020 | Electrophysiology of rat hippocampus Novelty and TMaze | https://doi.org/10.20350/digitalCSIC/12537 | digitalCSIC, 10.20350/digitalCSIC/12537 |

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
