## [Decision Letter]

**Acceptance summary:**

Hippocampal CA1 neurons play an important role in spatial navigation and memory, and their representations result from coordinated interactions between CA3 and enthorinal cortex that are structured by theta and gamma oscillations. Pérez-Montoyo et al. show that there are distinct sources of theta oscillations in different CA1 sub-layers, which arise from distinct hippocampal inputs and are uniquely associated with hippocampal gamma oscillations. Theta-gamma coupling and synchronization between theta generators increase with novelty and decision-making.

**Decision letter after peer review:**

Thank you for submitting your article "Different theta frameworks coexist in the hippocampus and are coordinated during memory-guided and novelty tasks" for consideration by *eLife*. Your article has been reviewed by four peer reviewers, one of whom is a member of our Board of Reviewing Editors, and the evaluation has been overseen by Laura Colgin as the Senior Editor. The following individual involved in review of your submission has agreed to reveal their identity: Antonio Fernandez-Ruiz (Reviewer #2).

The reviewers have discussed the reviews with one another and the Reviewing Editor has drafted this decision to help you prepare a revised submission.

Summary:

In this manuscript, Lopez-Madrona et al. identified different current sources of theta oscillations in local field potential recordings across dorsal hippocampus and characterized these different "theta-frameworks." Importantly, they showed that at least one of these theta-frameworks, the component with a source in the stratum radiatum (Sch-IC), is independent of the others by manipulating neurons in its source and only affecting the Sch-IC theta. Furthermore, the authors show that theta-gamma coupling and theta synchronization increase with novelty and decision-making.

Appreciation:

The reviewers in general made positive comments and considered that the work is a valuable addition to the field including many interesting analysis approaches, although some remarks were made about the novelty of the findings that require extensive revisions.

Revision:

1) Reviewers wish for further discussion on several points:

i) The authors showed that behavioral (speed) and cognitive (novelty, memory) factors also influence theta synchronization. It would be useful to more specifically address in the Discussion the interrelation of these factors with the activity of the different neuronal populations (CA3, entorhinal) that generate theta and gamma oscillations. Do the authors think that each theta and gamma generator is independently modulated or that is the global brain state that determines the "synchronization mode" in the whole network?

ii) The present results suggest that not only power and coherence but also theta frequency can be independently regulated for the different generators. That interpretation fits with other results in the paper and with the emergent view in the field of theta rhythm as a constellation of weakly-coupled local oscillators rather than a monolithic global rhythm. However, it challenges classical theories of theta generation. It would be interesting to discuss these aspects.

iii) Prior work using ICA, extracted rad-IC (slow gamma), LM-IC (medium gamma) and Pyr-IC (fast gamma, Fernández-Ruiz et al., 2017; Schomburg et al., 2014). The authors need to explain what is different or the same between Pyr-ICs in those works and PP-IC in this manuscript.

2) Independence of theta and gamma:

i) Reviewers pointed out that strong concerns have been raised by other authors w.r.t. to the existence of slow gamma oscillations in CA1. However, most of those concerns do not refer as much to this paper in particular as to the whole field. The existence of independent and distinguishable gamma oscillations in the hippocampus has been extensively demonstrated, notably by the Colgin, Klausberger and Buzsaki groups among many others. In general, reviewers consider the present results are in good agreement with most of the previous literature.

ii) It would be useful though to acknowledge the influence of factors, pointed by that reviewer, such as theta asymmetry and harmonics, type of analysis employed to separate gamma sub-bands, limitations of directionality measures, etc.

iii) The authors made some arguments like: "However, in this scenario, both the theta and gamma oscillations are components of the same signal and would be perfectly coupled." The argument is a bit limited.

iv) Further analyses are recommended to address this point: regress gamma power against theta power, speed and theta asymmetry (as it is known that this asymmetry is certainly the origin of increase power at higher harmonics). This would provide convincing argument that ICA makes it possible to separate independent gamma and theta signals.

v) Overall the authors should acknowledge this limitation and consider that carefully when they conclude that gamma leads theta.

3) Novelty:

i) The novelty and significance of the findings is not sufficiently clear. On the one hand, reviewers applauded the many new analysis approaches that make the paper a valuable contribution. On the other hand they commented that many, but not all, of the findings are very similar to prior reports, especially recently reports separating different types of theta cycles e.g. Dvorak et al., 2018, Lopes-dos-Santos et al., 2018, and Zhang et al., 2019. In light of these reports, the authors need to make much more clear what of their work is novel and what is either validation of prior work or replicating prior work to validate their approach. A key novel finding for example is that the authors can independently manipulate the Sch-IC theta pathway, because it shows conclusively that these different theta-frameworks are independent. Another main novel result is that theta itself, not just gamma therein, can shift in its coherence across subregions. Reviewers also liked the CFD analyses and the suggestion that theta is, somehow, the envelop of gamma "bursts".

4) Theta synchrony quantification:

The authors used the inter-cycle phase cluster index to quantify regional theta synchrony, which has some limitations. From the methods, ICPC was essentially phase locking between two ICs in a certain number of neighboring theta cycles (3 in the manuscript). This method might fail if the synchrony changed dynamically within two or one or even less than a theta cycle. Rapid theta dynamic are possible in light of recent work (Dvorak et al., 2018; Lopes-dos-Santos et al., 2018; Zhang et al., 2019), which were not discussed or mentioned in this manuscript. Using fewer theta cycles to calculate ICPC may be too unreliable, and using more theta cycles would have higher reliability but poorer time resolution. The authors needs to discuss this limitation in the way they measure and define regional synchrony.

5) Directionality analysis:

In the directionality analysis, gamma amplitude always led theta phase (Figure 5D), but it is unclear whether such consistency arose from the hippocampus or from the methods applied specifically to theta-gamma coupling. Is there control analyses that can ferret this out? Is there evidence in previous literature reporting slow oscillations leads fast oscillations in phase-amplitude coupling using this same analysis in another brain region? That would show this result is not just an artifact of this analysis method. Furthermore, it seems that PP-IC shows strongest lead of gamma (Figure 5D), what does this imply?

6) Optogenetics experiment:

i) As mentioned above, the optogenetic manipulation to show different theta-frameworks are independent is a very significant finding, however the experiments were only done in 3 animals. How consistent is the effect across animals and across recordings?

ii) Reviewers commented that the effect of inhibiting CA3 is small in terms of effect size – perhaps because the different ICA components still contain mixtures from multiple regions? Why was the equivalent experiment not done for the EC?

7) Questions about theta synchrony:

i) Is high theta synchrony simply due to high theta power? Are periods of low theta synchrony low theta power or messier theta cycles?

ii) During high synchrony periods, is the theta-coupled gamma band more synchronous as well? This is related to the question: can theta and gamma synchrony really be decoupled in the hippocampus?

8) Reviewers discussed the validity of the ICA analysis:

Some reviewers commented that ICA is designed find statistically independent components. Hence it seems perhaps confusing to then study interactions between different ICA components. The fact that there is synchronization among independent components could be highly confusing to some readers. The authors should discuss this and provide references for the validity of their ICA approach. Reviewers recommend including a diagram to help people understand they are performing ICA over spatially segregated LFPs and then performing temporal correlations between those spatially segregated components.

---

## [Author Response]

Revision:1) Reviewers wish for further discussion on several points:i) The authors showed that behavioral (speed) and cognitive (novelty, memory) factors also influence theta synchronization. It would be useful to more specifically address in the Discussion the interrelation of these factors with the activity of the different neuronal populations (CA3, entorhinal) that generate theta and gamma oscillations. Do the authors think that each theta and gamma generator is independently modulated or that is the global brain state that determines the "synchronization mode" in the whole network?

The main conclusion of our work is the existence of different theta frameworks in the hippocampus whose dynamics can be regulated independently. We show conclusively by optogenetic manipulation of CA3 PV+ interneurons that, indeed, the Schaffer generator can be modulated independently from the lacunoso-moleculare generator and the molecular layer generator of the DG. This independence, and the variable synchronization that we found between theta generators at different moments of the behaviors studied, allowed us to suggest a mechanism based on the coupling between these generators to integrate or segregate computations in the hippocampus. Furthermore, we hypothesized, based on our cross-frequency directionality analysis, that local and layer-specific gamma activities, reflecting locally specific excitation-inhibition interactions (Fernández-Ruiz et al., 2017; Lasztóczi and Klausberger, 2016, 2014; Somogyi et al., 2014), determine the phase of the corresponding theta oscillations, providing the mechanism to coordinate them. Therefore, we argue that generators are modulated independently and locally. Indeed, we show instances in which the three characterized theta generators are strongly coupled, others in which only two of them are strongly coupled or moments in which the three generators are weakly coupled. In turn, we speculate, these hippocampal theta coupling states will be associated with distinct brain-wide network states. In support of this view, we found specific behavioral/cognitive functions associated with different states of between-framework theta synchronization.

We have extended the text to make these points clearer and included the afferent regions (CA3 and EC) in the discussion of the behavioral and cognitive factors.

ii) The present results suggest that not only power and coherence but also theta frequency can be independently regulated for the different generators. That interpretation fits with other results in the paper and with the emergent view in the field of theta rhythm as a constellation of weakly-coupled local oscillators rather than a monolithic global rhythm. However, it challenges classical theories of theta generation. It would be interesting to discuss these aspects.

This is an important point that reinforces the idea of independent theta frameworks operating in the hippocampus and suggest that a mechanism for coupling/decoupling them is by fine-tuning their frequency. As mentioned above, our data suggests a mechanism based on the synchronization of theta generators to integrate or segregate computations in the hippocampus. However, our proposal is fundamentally different from others based on the segregation of computations in the phase of the theta wave (Hasselmo et al., 2002), or in single theta cycles (Lopes-Dos-Santos et al., 2018; Zhang et al., 2019) in that those are based on the rapid alternation of computational modes between phases or cycles, respectively, but always of a unique theta framework. In contrast, our proposal contemplates parallel processing in cell assemblies receiving information from different theta frameworks, whose computations can proceed segregated, preserving their content, but also integrated, combining information. In the first type of model, different operational modes in the hippocampus (i.e. exploring vs. retrieval) compete and separate in theta cycles and/or phases. In our model, different computations coexist and proceed in parallel avoiding interference when theta frameworks are decoupled (i.e. navigating in known environments) or integrating when thetas are synchronized (i.e. identifying novelty in a known environment and updating the memory accordingly). However, the two models complement each other, since computations in each theta framework will likely vary in a cycle-by-cycle manner.

We argue in the manuscript that external rhythm generators in the EC and septum are still fundamental for the generation of theta activity in the hippocampus, but the fine tuning of the different theta generators is likely controlled locally by layer specific excitation/inhibition interactions reflected in band-specific and theta-nested gamma oscillations. We have further discussed this point in the manuscript (Discussion, section “Independent theta frameworks”).

iii) Prior work using ICA, extracted rad-IC (slow gamma), LM-IC (medium gamma) and Pyr-IC (fast gamma, Fernández-Ruiz et al., 2017; Schomburg et al., 2014). The authors need to explain what is different or the same between Pyr-ICs in those works and PP-IC in this manuscript.

Pyr-IC and PP-IC are two different generators. The first has a current source with a peak at the pyramidal cell layer of CA1 and it has been suggested to represent the firing of the principal cells in CA1 (Schomburg et al., 2014). The second is originated by currents with peaks located in the mid-molecular layer of the DG and it has been identified as the excitatory input from the perforant pathway to the granule cells (Benito et al., 2014). The mentioned studies (Fernández-Ruiz et al., 2017; Schomburg et al., 2014) did not identified the PP-IC because the extraction of current generators requires a full coverage of the involved regions (Benito et al., 2014; Martín-Vázquez et al., 2016) and their electrodes were not fully covering the DG. In the present work, we employed electrodes with 32 recording sites separated by 100 µm which were able to cover CA1 and the DG. However, the tradeoff for covering both structures is that we do it with less density of recording points (one every 100 µm vs. 50 µm in Fernández-Ruiz et al., 2017; Schomburg et al., 2014). As a consequence, we do not have enough spatial resolution to identify the Pyr-IC generator, which has a very weak contribution to the total LFP variance. Therefore, we lose the information in Pyr-IC in favor of that in PP-IC.

2) Independence of theta and gamma:i) Reviewers pointed out that strong concerns have been raised by other authors w.r.t. to the existence of slow gamma oscillations in CA1. However, most of those concerns do not refer as much to this paper in particular as to the whole field. The existence of independent and distinguishable gamma oscillations in the hippocampus has been extensively demonstrated, notably by the Colgin, Klausberger and Buzsaki groups among many others. In general, reviewers consider the present results are in good agreement with most of the previous literature.ii) It would be useful though to acknowledge the influence of factors, pointed by that reviewer, such as theta asymmetry and harmonics, type of analysis employed to separate gamma sub-bands, limitations of directionality measures, etc.

We acknowledged in the manuscript these limitations and included in our analysis some measures to mitigate them. We have now included additional measures (i.e. the contribution of theta asymmetry to the low-gamma-theta CFC, see below and Figure 3—figure supplement 2), and extended the text correspondingly (section “Theta-gamma CFC reflects pathway-specific interactions”, second paragraph).

iii) The authors made some arguments like: "However, in this scenario, both the theta and gamma oscillations are components of the same signal and would be perfectly coupled." The argument is a bit limited.

This sentence was part of the argument regarding the use of CFD to evaluate the contribution of harmonics to the CFC, which has, indeed, some limitations. We have eliminated this sentence and acknowledge the limitations: section “Gamma oscillations consistently precede theta waves”, first paragraph.

iv) Further analyses are recommended to address this point: regress gamma power against theta power, speed and theta asymmetry (as it is known that this asymmetry is certainly the origin of increase power at higher harmonics). This would provide convincing argument that ICA makes it possible to separate independent gamma and theta signals.

We have performed the requested analysis, which is now included in the Figure 3—figure supplement 2 and main text (section “Theta-gamma CFC reflects pathway-specific interactions”, second paragraph). Briefly, we have computed a multiple linear regression for each IC-LFP, including theta power, speed and two metrics of theta asymmetry (rise-decay symmetry and peak-trough symmetry, (Cole and Voytek, 2018)) as factors to determine gamma power. Theta power was the main predictor in all IC-LFPs, as well as the speed in lm-IC. In all cases, the contribution of theta asymmetry to gamma power was negligible, supporting that theta-nested gamma bands have not its origin in theta harmonics.

v) Overall the authors should acknowledge this limitation and consider that carefully when they conclude that gamma leads theta.

We have reviewed the manuscript accordingly.

3) Novelty:i) The novelty and significance of the findings is not sufficiently clear. On the one hand, reviewers applauded the many new analysis approaches that make the paper a valuable contribution. On the other hand they commented that many, but not all, of the findings are very similar to prior reports, especially recently reports separating different types of theta cycles e.g. Dvorak et al., 2018, Lopes-dos-Santos et al., 2018, and Zhang et al., 2019. In light of these reports, the authors need to make much more clear what of their work is novel and what is either validation of prior work or replicating prior work to validate their approach. A key novel finding for example is that the authors can independently manipulate the Sch-IC theta pathway, because it shows conclusively that these different theta-frameworks are independent. Another main novel result is that theta itself, not just gamma therein, can shift in its coherence across subregions. Reviewers also liked the CFD analyses and the suggestion that theta is, somehow, the envelop of gamma "bursts".

We agree with the key novel points identified by the reviewers and have highlighted them throughout the text. We have also discussed our results in light of the recent reports separating different theta cycles (Dvorak et al., 2018, Lopes-dos-Santos et al., 2018, and Zhang et al., 2019), indicating the commonalities but also highlighting the important differences (one vs. multiple theta frameworks, as indicated above).

4) Theta synchrony quantification:The authors used the inter-cycle phase cluster index to quantify regional theta synchrony, which has some limitations. From the methods, ICPC was essentially phase locking between two ICs in a certain number of neighboring theta cycles (3 in the manuscript). This method might fail if the synchrony changed dynamically within two or one or even less than a theta cycle. Rapid theta dynamic are possible in light of recent work (Dvorak et al., 2018; Lopes-dos-Santos et al., 2018; Zhang et al., 2019), which were not discussed or mentioned in this manuscript. Using fewer theta cycles to calculate ICPC may be too unreliable, and using more theta cycles would have higher reliability but poorer time resolution. The authors needs to discuss this limitation in the way they measure and define regional synchrony.

We agree in the limitation of the ICPC and have acknowledged this point in the manuscript (section “Different theta frameworks coexist in the dorsal hippocampus”, third paragraph). However, in order to provide a finer time scale of the changes in theta synchronization, we analyzed the variability of the ICPC across time by comparing the value of a given cycle to that of the previous ones (Figure 2G). We found that the coupling strength between consecutive cycles spreads on a time scale in the order of one second (0.75 s), thus expecting dynamical changes in the ICPC in this time scale. This methodology allowed the characterization of the temporal synchronization between theta generators using ICPC with a time resolution of 1 theta cycle. Future studies combining the multiple theta-framework concept introduced here, with the single theta cycle analysis recently developed (Dvorak et al., 2018; Lopes-Dos-Santos et al., 2018; Zhang et al., 2019) will provide comprehensive analysis of hippocampal theta computations.

5) Directionality analysis:In the directionality analysis, gamma amplitude always led theta phase (Figure 5D), but it is unclear whether such consistency arose from the hippocampus or from the methods applied specifically to theta-gamma coupling. Is there control analyses that can ferret this out? Is there evidence in previous literature reporting slow oscillations leads fast oscillations in phase-amplitude coupling using this same analysis in another brain region? That would show this result is not just an artifact of this analysis method.

Yes, there is evidence of slow oscillations leading to fast ones using CFD. First, in the work where the method was proposed, Jiang et al., 2015 showed its effectiveness in models, describing scenarios with the slow oscillation leading the fast, the fast leading the slow and situations with cross-frequency coupling but without directionality. Examples of these scenarios can be found in real brain signals with the same methodology. In Helfrich et al., 2019, 2018, phases of cortical slow oscillations significantly predicted sleep spindle amplitudes in humans, and in Zheng et al., 2017 low-frequency activity from amygdala preceded hippocampus gamma. In the opposite case, Jiang et al., 2015 found that the power envelope of gamma oscillations drives the phase of slower oscillations in the α band using electrocorticography in humans.

Furthermore, it seems that PP-IC shows strongest lead of gamma (Figure 5D), what does this imply?

The differences in the CFD values can be attributed to the different strengths of the theta-gamma coupling in each IC. As shown in Figure 3, the gamma oscillations in PP-IC show a stronger CFC to the theta rhythm, which would imply also a higher CFD in absolute value, as it is the case.

6) Optogenetics experiment:i) As mentioned above, the optogenetic manipulation to show different theta-frameworks are independent is a very significant finding, however the experiments were only done in 3 animals. How consistent is the effect across animals and across recordings?

We have included two new panels to show that the results were highly consistent across animals (black lines, Figure 2E). Moreover, we have included a representative example of the effect of light stimulation on Sch-IC time course in one trial (Figure 2D).

ii) Reviewers commented that the effect of inhibiting CA3 is small in terms of effect size – perhaps because the different ICA components still contain mixtures from multiple regions? Why was the equivalent experiment not done for the EC?

The optogenetic experiment was performed using optic fibers of 100 μm radius 0.66 NA, implanted over the CA3 region. Using our LED system, light irradiance at the tip of the fiber was 50 mW/mm2. Taking into account light transmission in brain tissue (Yizhar et al., 2011), irradiance decayed to 10 mW/mm2 at 100 μm from the fiber’s tip, and below 1 mW/mm2 at 500 μm. From previous works, we know that wild-type ChR2 at typical expression levels and illuminated with 473 nm light at power densities between 1-5 mW/mm2 is able to elicit action potential firing (Boyden et al., 2005; Yizhar et al., 2011). Taken together, we estimate that our optogenetic manipulation activated PV+ interneurons roughly in a cylinder of 100 μm radius and 500 μm length, which represents a small proportion of the CA3 region providing the Shaffer collateral input to the adjacent CA1. Therefore, although some residual cross-contamination between ICs cannot be fully discarded, a more likely explanation of the effect size of the optogenetic experiment is the quantitative mismatch between the large number of Schaffer inputs recorded by the electrode implanted in CA1, and the comparably modest number of fibers under the modulatory effect of blue-light illumination.

Regarding the EC experiment, the possibilities to perform a clean experiment to specifically and selectively manipulate layer II or layer III of the EC are very limited in rats. Furthermore, there are anatomical constraints that preclude the possibility of a specific manipulation of the PP input to the DG since layer II output also contacts CA3. Due to these limitations, we focused our optogenetic experiment on the Sch generator.

7) Questions about theta synchrony:i) Is high theta synchrony simply due to high theta power? Are periods of low theta synchrony low theta power or messier theta cycles?

These two questions were accounted for in our analysis. For the first one, it is expected that low theta power cycles may result in less accurate measurements of its phase and, therefore, spurious periods of low synchronization. Knowing this fact, however, we limited our analysis to cycles with good signal-to-noise ratio based on the ratio between theta and delta power (Jackson et al., 2006; Ólafsdóttir et al., 2017), and validated mathematically this threshold (Materials and methods and Figure 2—figure supplement 2). In this conditions, the error in the estimation of the phase is below 1 % of the theta cycle and, therefore, lower synchrony epochs cannot be attributed to inaccurate phase estimation. Furthermore, although theta power and synchrony positively correlate (Figure 4B), we tested that theta power is not the only factor affecting to theta synchronization (Figure 4C). Most importantly, and answering the second question, there are epochs with poor synchronization but high theta power (Figure 4D). Finally, certain level of correlation between theta power and synchrony is expected since more synchronized currents in the neuronal population would generate extracellular field potentials of larger magnitude. Therefore, although we found a correlation between the increase of theta power and synchrony, we ensured that the differences in theta synchronization (1) are not an artefact due to low theta power, (2) are not solely explain by theta power and (3) we consistently found epochs of high synchronization but low theta power.

ii) During high synchrony periods, is the theta-coupled gamma band more synchronous as well? This is related to the question: can theta and gamma synchrony really be decoupled in the hippocampus?

Theta-coupled gamma activity in each IC-LFP differs from the others in its frequency (Figure 3B), and the phase of the theta cycle to which they are nested (Figure 3C). Therefore, a synchrony between these activities cannot be achieved straightforward. Please note that, in contrast to previous works comparing gamma coherence between separated regions, like low-gamma coherence between CA3 and CA1 and medium-gamma coherence between EC and CA1 (Colgin et al., 2009), here we are not considering CA3 nor EC signals per se, but the effect of their efferences on the activity of CA1 and DG. What our analyses revealed is that gamma activity is better coupled to the theta rhythm when they are synchronized across regions (Figure 4B). In this scenario, it is expected a higher coherence between the envelopes of the different gamma oscillations under theta synchronization, but not between the oscillations themselves (see Author response image 1). Regarding the question about if gamma and theta synchrony can be decoupled, our data suggest that both phenomena are related but relatively independent. The theta-gamma coupling in each IC-LFP can be high even in states of low theta synchronization (Figure 4B), an indication of gamma processes nested in theta cycles but in absence of theta synchronization. Moreover, it should be noted that the presence of theta activity is not a prerequisite for gamma oscillations (Csicsvari et al., 2003).

**Author response image 1. sa2fig1:** Spectral coherence analysis between IC-LFPs. Theta synchronization states are separated based on ICPC (color legend). Note the low coherence at gamma frequencies between IC-LFPs.

8) Reviewers discussed the validity of the ICA analysis:Some reviewers commented that ICA is designed find statistically independent components. Hence it seems perhaps confusing to then study interactions between different ICA components. The fact that there is synchronization among independent components could be highly confusing to some readers. The authors should discuss this and provide references for the validity of their ICA approach. Reviewers recommend to include a diagram to help people understand they are performing ICA over spatially segregated LFPs and then performing temporal correlations between those spatially segregated components.

We have included more references in the main text with validation and examples of uses of our ICA approach (section “Pathway-specific theta and gamma oscillations”, first paragraph). We have also specified in the main text that it is only necessary a small amount of time independency to separate sources with ICA (section “Pathway-specific theta and gamma oscillations”, third paragraph), and kept the extended description of the methodology in the Materials and methods section. Moreover, we have included a new diagram to facilitate the understanding of the ICA approach (Figure 2—figure supplement 1).